# Neural general circulation models for weather and climate

Dmitrii Kochkov[1,6 ✉], Janni Yuval[1,6 ✉], Ian Langmore[1,6], Peter Norgaard[1,6], Jamie Smith[1,6], Griffin Mooers[1], Milan Klöwer[2], James Lottes[1], Stephan Rasp[1], Peter Düben[3], Sam Hatfield[3], Peter Battaglia[4], Alvaro Sanchez-Gonzalez[4], Matthew Willson[4], Michael P. Brenner[1,5] & Stephan Hoyer[1,6 ✉]

General circulation models (GCMs) are the foundation of weather and climate prediction[1,2]. GCMs are physics-based simulators that combine a numerical solver for large-scale dynamics with tuned representations for small-scale processes such as cloud formation. Recently, machine-learning models trained on reanalysis data have achieved comparable or better skill than GCMs for deterministic weather forecasting[3,4]. However, these models have not demonstrated improved ensemble forecasts, or shown sufficient stability for long-term weather and climate simulations. Here we present a GCM that combines a differentiable solver for atmospheric dynamics with machine-learning components and show that it can generate forecasts of deterministic weather, ensemble weather and climate on par with the best machine-learning and physics-based methods. NeuralGCM is competitive with machine-learning models for one- to ten-day forecasts, and with the European Centre for Medium-Range Weather Forecasts ensemble prediction for one- to fifteen-day forecasts. With prescribed sea surface temperature, NeuralGCM can accurately track climate metrics for multiple decades, and climate forecasts with 140-kilometre resolution show emergent phenomena such as realistic frequency and trajectories of tropical cyclones. For both weather and climate, our approach offers orders of magnitude computational savings over conventional GCMs, although our model does not extrapolate to substantially different future climates. Our results show that end-to-end deep learning is compatible with tasks performed by conventional GCMs and can enhance the large-scale physical simulations that are essential for understanding and predicting the Earth system.

Solving the equations for Earth's atmosphere with general circulation models (GCMs) is the basis of weather and climate prediction[1,2]. Over the past 70 years, GCMs have been steadily improved with better numerical methods and more detailed physical models, while exploiting faster computers to run at higher resolution. Inside GCMs, the unresolved physical processes such as clouds, radiation and precipitation are represented by semi-empirical parameterizations. Tuning GCMs to match historical data remains a manual process[5], and GCMs retain many persistent errors and biases[6–8]. The difficulty of reducing uncertainty in long-term climate projections[9] and estimating distributions of extreme weather events[10] presents major challenges for climate mitigation and adaptation[11].

Recent advances in machine learning have presented an alternative for weather forecasting[3,4,12,13]. These models rely solely on machine-learning techniques, using roughly 40 years of historical data from the European Center for Medium-Range Weather Forecasts (ECMWF) reanalysis v5 (ERA5)[14] for model training and forecast initialization. Machine-learning methods have been remarkably successful,

demonstrating state-of-the-art deterministic forecasts for 1- to 10-day weather prediction at a fraction of the computational cost of traditional models[3,4]. Machine-learning atmospheric models also require considerably less code, for example GraphCast[3] has 5,417 lines versus 376,578 lines for the National Oceanic and Atmospheric Administration's FV3 atmospheric model[15] (see Supplementary Information section A for details).

Nevertheless, machine-learning approaches have noteworthy limitations compared with GCMs. Existing machine-learning models have focused on deterministic prediction, and surpass deterministic numerical weather prediction in terms of the aggregate metrics for which they are trained[3,4]. However, they do not produce calibrated uncertainty estimates[4], which is essential for useful weather forecasts[1]. Deterministic machine-learning models using a mean-squared-error loss are rewarded for averaging over uncertainty, producing unrealistically blurry predictions when optimized for multi-day forecasts[3,13]. Unlike physical models, machine-learning models misrepresent derived (diagnostic) variables such as geostrophic wind[16]. Furthermore,

[1]Google Research, Mountain View, CA, USA. [2]Earth, Atmospheric and Planetary Sciences, Massachusetts Institute of Technology, Cambridge, MA, USA. [3]European Centre for Medium-Range Weather Forecasts, Reading, UK. [4]Google DeepMind, London, UK. [5]School of Engineering and Applied Sciences, Harvard University, Cambridge, MA, USA. [6]These authors contributed equally: Dmitrii Kochkov, Janni Yuval, Ian Langmore, Peter Norgaard, Jamie Smith, Stephan Hoyer. ✉e-mail: dkochkov@google.com; janniyuval@google.com; shoyer@google.com

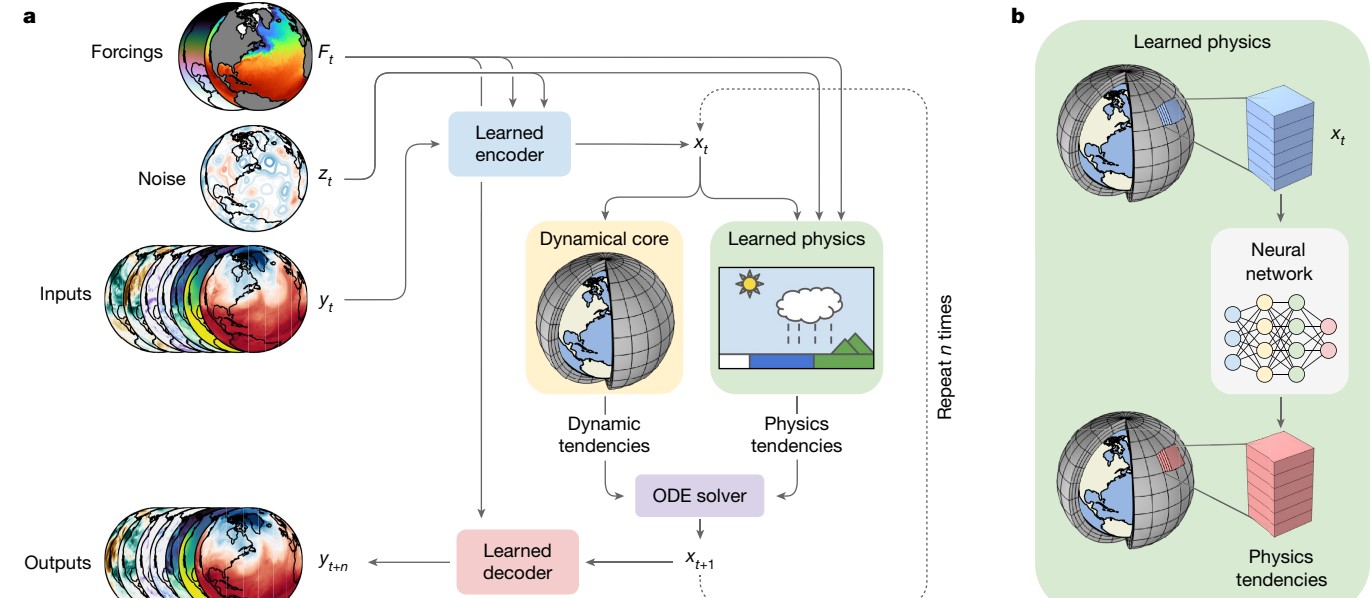

**Fig. 1 | Structure of the NeuralGCM model. a**, Overall model structure, showing how forcings $F_t$, noise $z_t$ (for stochastic models) and inputs $y_t$ are encoded into the model state $x_t$. The model state is fed into the dynamical core, and alongside forcings and noise into the learned physics module. This produces tendencies (rates of change) used by an implicit–explicit ordinary differential equation (ODE) solver to advance the state in time. The new model state $x_{t+1}$ can then be fed back into another time step, or decoded into model predictions. **b**, The learned physics module, which feeds data for individual columns of the atmosphere into a neural network used to produce physics tendencies in that vertical column.

although there has been some success in using machine-learning approaches on longer timescales[17,18], these models have not demonstrated the ability to outperform existing GCMs.

Hybrid models that combine GCMs with machine learning are appealing because they build on the interpretability, extensibility and successful track record of traditional atmospheric models[19,20]. In the hybrid model approach, a machine-learning component replaces or corrects the traditional physical parameterizations of a GCM. Until now, the machine-learning component in such models has been trained 'offline', by learning parameterizations independently of their interaction with dynamics. These components are then inserted into an existing GCM. The lack of coupling between machine-learning components and the governing equations during training potentially causes serious problems, such as instability and climate drift[21]. So far, hybrid models have mostly been limited to idealized scenarios such as aquaplanets[22,23]. Under realistic conditions, machine-learning corrections have reduced some biases of very coarse GCMs[24–26], but performance remains considerably worse than state-of-the-art models.

Here we present NeuralGCM, a fully differentiable hybrid GCM of Earth's atmosphere. NeuralGCM is trained on forecasting up to 5-day weather trajectories sampled from ERA5. Differentiability enables end-to-end 'online training'[27], with machine-learning components optimized in the context of interactions with the governing equations for large-scale dynamics, which we find enables accurate and stable forecasts. NeuralGCM produces physically consistent forecasts with accuracy comparable to best-in-class models across a range of timescales, from 1- to 15-day weather to decadal climate prediction.

## Neural GCMs

A schematic of NeuralGCM is shown in Fig. 1. The two key components of NeuralGCM are a differentiable dynamical core for solving the discretized governing dynamical equations and a learned physics module that parameterizes physical processes with a neural network, described in full detail in Methods, Supplementary Information sections B and C, and Supplementary Table 1. The dynamical core simulates large-scale fluid motion and thermodynamics under the influence of gravity and the Coriolis force. The learned physics module (Supplementary Fig. 1) predicts the effect of unresolved processes, such as cloud formation, radiative transport, precipitation and subgrid-scale dynamics, on the simulated fields using a neural network.

The differentiable dynamical core in NeuralGCM allows an end-to-end training approach, whereby we advance the model multiple time steps before employing stochastic gradient descent to minimize discrepancies between model predictions and reanalysis (Supplementary Information section G.2). We gradually increase the rollout length from 6 hours to 5 days (Supplementary Information section G and Supplementary Table 5), which we found to be critical because our models are not accurate for multi-day prediction or stable for long rollouts early in training (Supplementary Information section H.6.2 and Supplementary Fig. 23). The extended back-propagation through hundreds of simulation steps enables our neural networks to take into account interactions between the learned physics and the dynamical core. We train deterministic and stochastic NeuralGCM models, each of which uses a distinct training protocol, described in full detail in Methods and Supplementary Table 4.

We train a range of NeuralGCM models at horizontal resolutions with grid spacing of 2.8°, 1.4° and 0.7° (Supplementary Fig. 7). We evaluate the performance of NeuralGCM at a range of timescales appropriate for weather forecasting and climate simulation. For weather, we compare against the best-in-class conventional physics-based weather models, ECMWF's high-resolution model (ECMWF-HRES) and ensemble prediction system (ECMWF-ENS), and two of the recent machine-learning-based approaches, GraphCast[3] and Pangu[4]. For climate, we compare against a global cloud-resolving model and Atmospheric Model Intercomparison Project (AMIP) runs.

## Medium-range weather forecasting

Our evaluation set-up focuses on quantifying accuracy and physical consistency, following WeatherBench2[12]. We regrid all forecasts to a 1.5° grid using conservative regridding, and average over all 732 forecasts

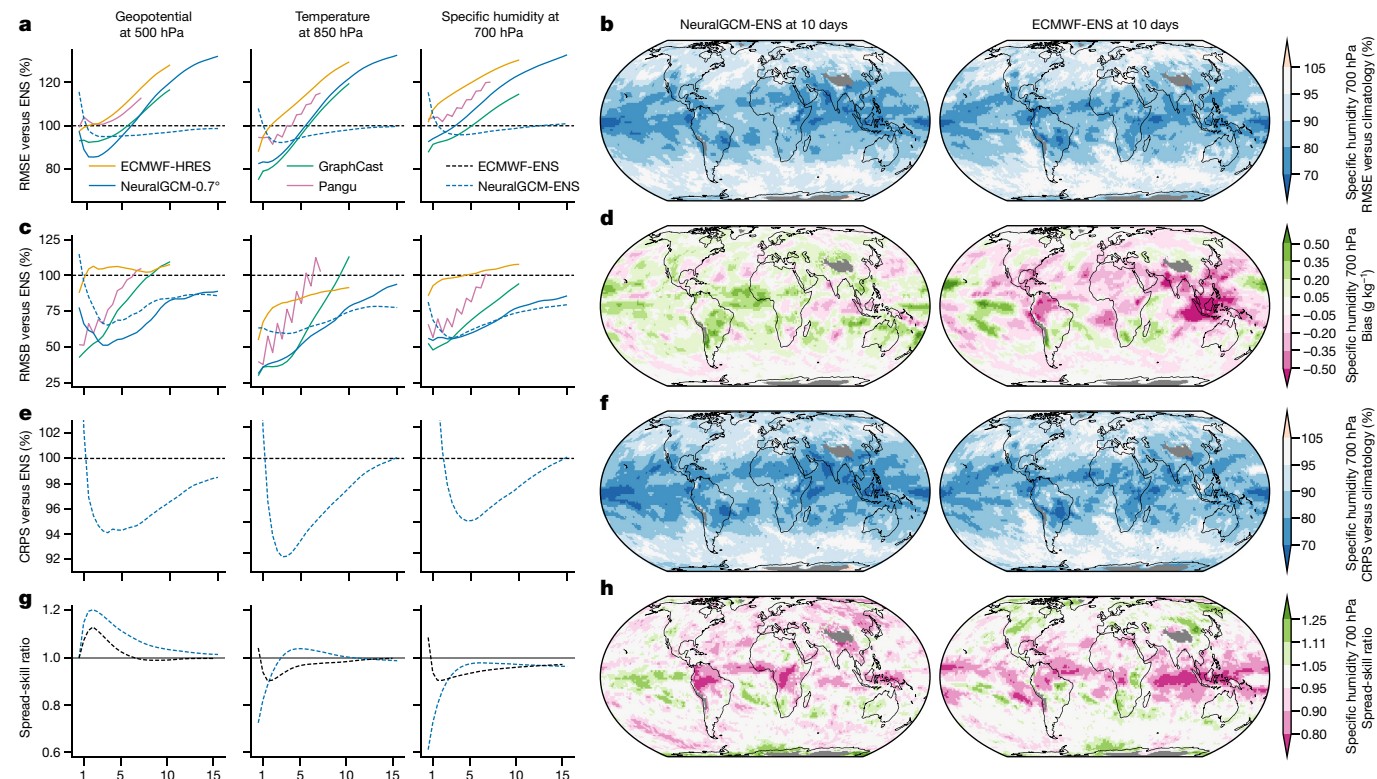

**Fig. 2 | Weather forecasting accuracy scores for deterministic and stochastic models. a,c**, RMSE (**a**) and RMSB (**c**) for ECMWF-ENS, ECMWF-HRES, NeuralGCM-0.7°, NeuralGCM-ENS, GraphCast[3] and Pangu[4] on headline WeatherBench2 variables, as a percentage of the error of ECMWF-ENS. Deterministic and stochastic models are shown in solid and dashed lines respectively. **e,g**, CRPS relative to ECMWF-ENS (**e**) and spread-skill ratio for the

ENS and NeuralGCM-ENS models (**g**). **b,d,f,h**, Spatial distributions of RMSE (**b**), bias (**d**), CRPS (**f**) and spread-skill ratio (**h**) for NeuralGCM-ENS and ECMWF-ENS models for 10-day forecasts of specific humidity at 700 hPa. Spatial plots of RMSE and CRPS show skill relative to a probabilistic climatology[12] with an ensemble member for each of the years 1990–2019. The grey areas indicate regions where climatological surface pressure on average is below 700 hPa.

made at noon and midnight UTC in the year 2020, which was held-out from training data for all machine-learning models. NeuralGCM, GraphCast and Pangu compare with ERA5 as the ground truth, whereas ECMWF-ENS and ECMWF-HRES compare with the ECMWF operational analysis (that is, HRES at 0-hour lead time), to avoid penalizing the operational forecasts for different biases than ERA5.

## Model accuracy

We use ECMWF's ensemble (ENS) model as a reference baseline as it achieves the best performance across the majority of lead times[12]. We assess accuracy using (1) root-mean-squared error (RMSE), (2) root-mean-squared bias (RMSB), (3) continuous ranked probability score (CRPS) and (4) spread-skill ratio, with the results shown in Fig. 2. We provide more in-depth evaluations including scorecards, metrics for additional variables and levels and maps in Extended Data Figs. 1 and 2, Supplementary Information section H and Supplementary Figs. 9–22.

Deterministic models that produce a single weather forecast for given initial conditions can be compared effectively using RMSE skill at short lead times. For the first 1–3 days, depending on the atmospheric variable, RMSE is minimized by forecasts that accurately track the evolution of weather patterns. At this timescale we find that NeuralGCM-0.7° and GraphCast achieve best results, with slight variations across different variables (Fig. 2a). At longer lead times, RMSE rapidly increases owing to chaotic divergence of nearby weather trajectories, making RMSE less informative for deterministic models. RMSB calculates persistent errors over time, which provides an indication of how models would perform at much longer lead times. Here NeuralGCM models also compare favourably against previous approaches (Fig. 2c), with notably much less bias for specific humidity in the tropics (Fig. 2d).

Ensembles are essential for capturing intrinsic uncertainty of weather forecasts, especially at longer lead times. Beyond about 7 days, the ensemble means of ECMWF-ENS and NeuralGCM-ENS forecasts have considerably lower RMSE than the deterministic models, indicating that these models better capture the average of possible weather. A better metric for ensemble models is CRPS, which is a proper scoring rule that is sensitive to full marginal probability distributions[28]. Our stochastic model (NeuralGCM-ENS) running at 1.4° resolution has lower error compared with ECMWF-ENS across almost all variables, lead times and vertical levels for ensemble-mean RMSE, RSMB and CRPS (Fig. 2a,c,e and Supplementary Information section H), with similar spatial patterns of skill (Fig. 2b,f). Like ECMWF-ENS, NeuralGCM-ENS has a spread-skill ratio of approximately one (Fig. 2d), which is a necessary condition for calibrated forecasts[29].

## Case study

An important characteristic of forecasts is their resemblance to realistic weather patterns. Figure 3 shows a case study that illustrates the performance of NeuralGCM on three types of important weather phenomenon: tropical cyclones, atmospheric rivers and the Intertropical Convergence Zone. Figure 3a shows that all the machine-learning models make significantly blurrier forecasts than the source data ERA5 and physics-based ECMWF-HRES forecast, but NeuralCGM-0.7° outperforms the pure machine-learning models, despite its coarser resolution (0.7° versus 0.25° for GraphCast and Pangu). Blurry forecasts correspond to physically inconsistent atmospheric conditions and misrepresent extreme weather. Similar trends hold for other derived variables of meteorological interest (Supplementary Information section H.2). Ensemble-mean predictions, from both NeuralGCM and ECMWF, are

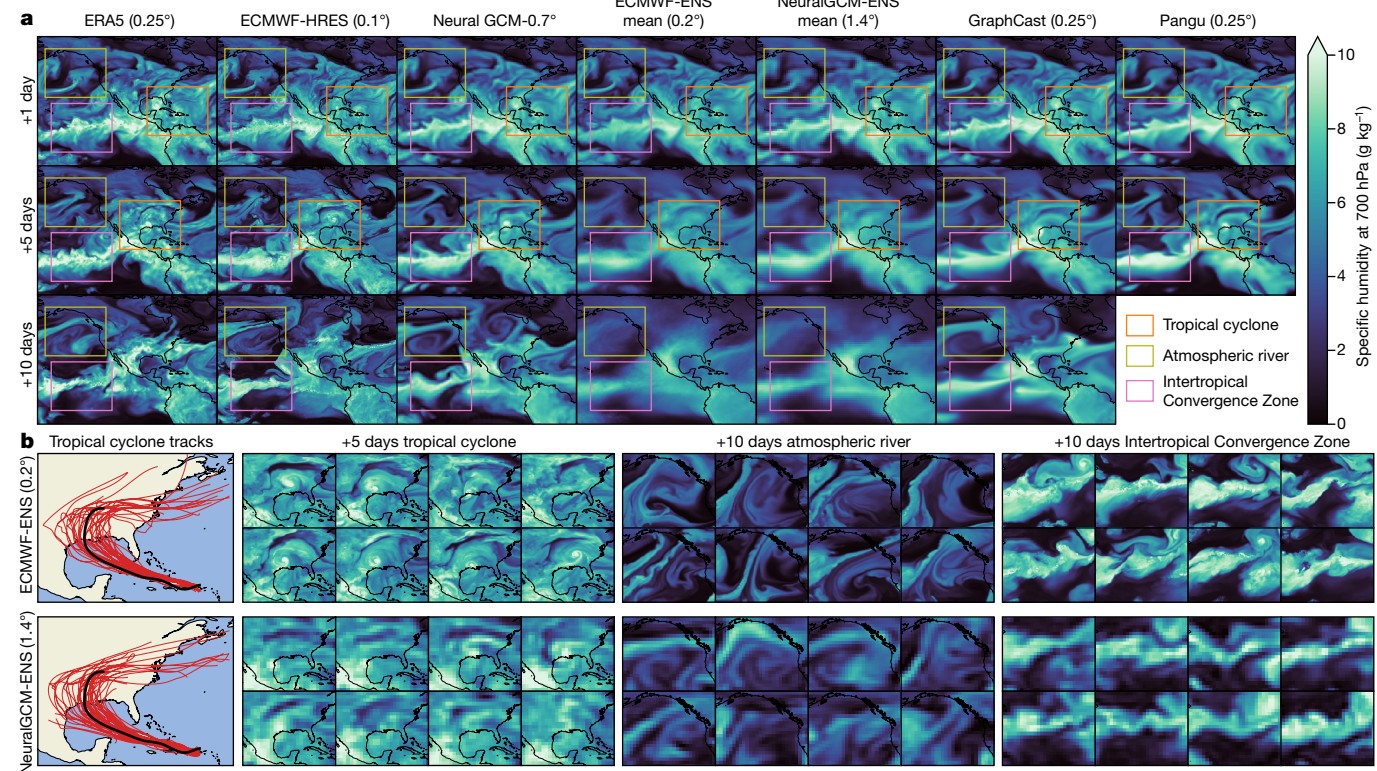

**Fig. 3 | Case study of a medium-range weather forecast.** All forecasts are initialized at 2020-08-22T12z, chosen to highlight Hurricane Laura, the most damaging Atlantic hurricane of 2020. **a**, Specific humidity at 700 hPa for 1-day, 5-day and 10-day forecasts over North America and the Northeast Pacific Ocean from ERA5[14], ECMWF-HRES, NeuralGCM-0.7°, ECMWF-ENS (mean), NeuralGCM-ENS (mean), GraphCast[3] and Pangu[4]. **b**, Forecasts from individual ensemble members from ECMWF-ENS and NeuralGCM-ENS over regions of interest, including predicted tracks of Hurricane Laura from each of the 50 ensemble members (Supplementary Information section I.2). The track from ERA5 is plotted in black.

closer to ERA5 in an average sense, and thus are inherently smooth at long lead times. In contrast, as shown in Fig. 3 and in Supplementary Information section H.3, individual realizations from the ECMWF and NeuralGCM ensembles remain sharp, even at long lead times. Like ECMWF-ENS, NeuralGCM-ENS produces a statistically representative range of future weather scenarios for each weather phenomenon, despite its eight-times-coarser resolution.

### Spectra

We can quantify the blurriness of different forecast models via their power spectra. Supplementary Figs. 17 and 18 show that the power spectra of NeuralCGM-0.7° is consistently closer to ERA5 than the other machine-learning forecast methods, but is still blurrier than ECMWF's physical forecasts. The spectra of NeuralGCM forecasts is also roughly constant over the forecast period, in stark contrast to GraphCast, which worsens with lead time. The spectrum of NeuralGCM becomes more accurate with increased resolution (Supplementary Fig. 22), which suggests the potential for further improvements of NeuralGCM models trained at higher resolutions.

### Water budget

In NeuralGCM, advection is handled by the dynamical core, while the machine-learning parameterization models local processes within vertical columns of the atmosphere. Thus, unlike pure machine-learning methods, local sources and sinks can be isolated from tendencies owing to horizontal transport and other resolved dynamics (Supplementary Fig. 3). This makes our results more interpretable and facilitates the diagnosis of the water budget. Specifically, we diagnose precipitation minus evaporation (Supplementary Information section H.5) rather than directly predicting these as in machine-learning-based approaches[3].

For short weather forecasts, the mean of precipitation minus evaporation has a realistic spatial distribution that is very close to ERA5 data (Extended Data Fig. 4c–e). The precipitation-minus-evaporation rate distribution of NeuralGCM-0.7° closely matches the ERA5 distribution in the extratropics (Extended Data Fig. 4b), although it underestimates extreme events in the tropics (Extended Data Fig. 4a). It is noted that the current version of NeuralGCM directly predicts tendencies for an atmospheric column, and thus cannot distinguish between precipitation and evaporation.

### Geostrophic wind balance

We examined the extent to which NeuralGCM, GraphCast and ECMWF-HRES capture the geostrophic wind balance, the near-equilibrium between the dominant forces that drive large-scale dynamics in the mid-latitudes[30]. A recent study[16] highlighted that Pangu misrepresents the vertical structure of the geostrophic and ageostrophic winds and noted a deterioration at longer lead times. Similarly, we observe that GraphCast shows an error that worsens with lead time. In contrast, NeuralGCM more accurately depicts the vertical structure of the geostrophic and ageostrophic winds, as well as their ratio, compared with GraphCast across various rollouts, when compared against ERA5 data (Extended Data Fig. 3). However, ECMWF-HRES still shows a slightly closer alignment to ERA5 data than NeuralGCM does. Within Neural-GCM, the representation of the geostrophic wind's vertical structure only slightly degrades in the initial few days, showing no noticeable changes thereafter, particularly beyond day 5.

### Generalizing to unseen data

Physically consistent weather models should still perform well for weather conditions for which they were not trained. We expect that

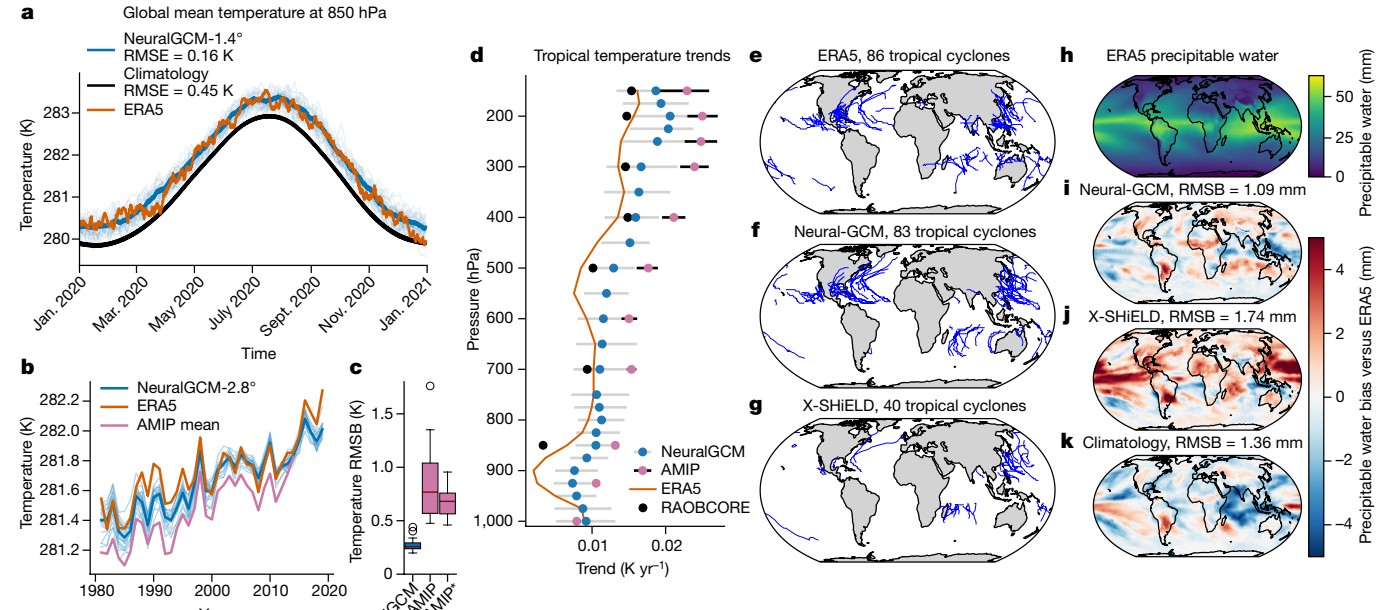

**Fig. 4 | Simulation of climate with NeuralGCM. a**, Global mean temperature for ERA5[14] (orange), 1990–2019 climatology (black) and NeuralGCM-1.4° (blue) for 2020 using 35 simulations initialized every 10 days during 2019 (thick line, ensemble mean; thin lines, different initial conditions). **b**, Yearly global mean temperature for ERA5 (orange), mean over 22 CMIP6 AMIP experiments[34] (violet; model details are in Supplementary Information section I.3) and NeuralGCM-2.8° for 22 AMIP-like simulations with prescribed SST initialized every 10 days during 1980 (thick line, ensemble mean; thin lines, different initial conditions). **c**, The RMSB of the 850-hPa temperature averaged between 1981 and 2014 for 22 NeuralGCM-2.8° AMIP runs (labelled NGCM), 22 CMIP6 AMIP experiments (labelled AMIP) and debiased 22 CMIP6 AMIP experiments (labelled AMIP*; bias was removed by removing the 850-hPa global temperature bias). In the box plots, the red line represents the median. The box delineates the first to third quartiles; the whiskers extend to 1.5 times the interquartile range (Q1 − 1.5IQR and Q3 + 1.5IQR), and outliers are shown as individual dots. **d**, Vertical profiles of tropical (20° S–20° N) temperature trends for 1981–2014. Orange, ERA5; black dots, Radiosonde Observation Correction using Reanalyses (RAOBCORE)[41]; blue dots, mean trends for NeuralGCM; purple dots, mean trends from CMIP6 AMIP runs (grey and black whiskers, 25th and 75th percentiles for NeuralGCM and CMIP6 AMIP runs, respectively). **e**–**g**, Tropical cyclone tracks for ERA5 (**e**), NeuralGCM-1.4° (**f**) and X-SHiELD[31] (**g**). **h**–**k**, Mean precipitable water for ERA5 (**h**) and the precipitable water bias in NeuralGCM-1.4° (**i**), initialized 90 days before mid-January 2020 similarly to X-SHiELD, X-SHiELD (**j**) and climatology (**k**; averaged between 1990 and 2019). In **d**–**i**, quantities are calculated between mid-January 2020 and mid-January 2021 and all models were regridded to a 256 × 128 Gaussian grid before computation and tracking.

## Climate simulations

Although our deterministic NeuralGCM models are trained to predict weather up to 3 days ahead, they are generally capable of simulating the atmosphere far beyond medium-range weather timescales. For extended climate simulations, we prescribe historical sea surface temperature (SST) and sea-ice concentration. These simulations feature many emergent phenomena of the atmosphere on timescales from months to decades.

For climate simulations with NeuralGCM, we use 2.8° and 1.4° deterministic models, which are relatively inexpensive to train (Supplementary Information section G.7) and allow us to explore a larger parameter space to find stable models. Previous studies found that running extended simulations with hybrid models is challenging due to numerical instabilities and climate drift[21]. To quantify stability in our selected models, we run multiple initial conditions and report how many of them finish without instability.

### Seasonal cycle and emergent phenomena

To assess the capability of NeuralGCM to simulate various aspects of the seasonal cycle, we run 2-year simulations with NeuralGCM-1.4°. for 37 different initial conditions spaced every 10 days for the year 2019. Out of these 37 initial conditions, 35 successfully complete the full 2 years without instability; for case studies of instability, see Supplementary Information section H.7, and Supplementary Figs. 26 and 27. We compare results from NeuralGCM-1.4° for 2020 with ERA5 data and with outputs from the X-SHiELD global cloud-resolving model, which is coupled to an ocean model nudged towards reanalysis[31]. This X-SHiELD run has been used as a target for training machine-learning climate models[24]. For comparison, we evaluate models after regridding predictions to 1.4° resolution. This comparison slightly favours NeuralGCM because NeuralGCM was tuned to match ERA5, but the discrepancy between ERA5 and the actual atmosphere is small relative to model error.

Figure 4a shows the temporal variation of the global mean temperature to 2020, as captured by 35 simulations from NeuralGCM, in comparison with the ERA5 reanalysis and standard climatology benchmarks. The seasonality and variability of the global mean temperature from NeuralGCM are quantitatively similar to those observed in ERA5. The ensemble-mean temperature RMSE for NeuralGCM stands at 0.16 K when benchmarked against ERA5, which is a significant improvement over the climatology's RMSE of 0.45 K. We find that NeuralGCM

NeuralGCM may generalize better than machine-learning-only atmospheric models, because NeuralGCM employs neural networks that act locally in space, on individual vertical columns of the atmosphere. To explore this hypothesis, we compare versions of NeuralCGM-0.7° and GraphCast trained to 2017 on 5 years of weather forecasts beyond the training period (2018–2022) in Supplementary Fig. 36. Unlike Graph-Cast, NeuralGCM does not show a clear trend of increasing error when initialized further into the future from the training data. To extend this test beyond 5 years, we trained a NeuralGCM-2.8° model using only data before 2000, and tested its skill for over 21 unseen years (Supplementary Fig. 35).

accurately simulates the seasonal cycle, as evidenced by metrics such as the annual cycle of the global precipitable water (Supplementary Fig. 30a) and global total kinetic energy (Supplementary Fig. 30b). Furthermore, the model captures essential atmospheric dynamics, including the Hadley circulation and the zonal-mean zonal wind (Supplementary Fig. 28), as well as the spatial patterns of eddy kinetic energy in different seasons (Supplementary Fig. 31), and the distinctive seasonal behaviours of monsoon circulation (Supplementary Fig. 29; additional details are provided in Supplementary Information section I.1).

Next, we compare the annual biases of a single NeuralGCM realization with a single realization of X-SHiELD (the only one available), both initiated in mid-October 2019. We consider 19 January 2020 to 17 January 2021, the time frame for which X-SHiELD data are available. Global cloud-resolving models, such as X-SHiELD, are considered state of the art, especially for simulating the hydrological cycle, owing to their resolution being capable of resolving deep convection[32]. The annual bias in precipitable water for NeuralGCM (RMSE of 1.09 mm) is substantially smaller than the biases of both X-SHiELD (RMSE of 1.74 mm) and climatology (RMSE of 1.36 mm; Fig. 4i–k). Moreover, NeuralGCM shows a lower temperature bias in the upper and lower troposphere than X-SHiELD (Extended Data Fig. 6). We also indirectly compare precipitation bias in X-SHiELD with precipitation-minus-evaporation bias in NeuralGCM-1.4°, which shows slightly larger bias and grid-scale artefacts for NeuralGCM (Extended Data Fig. 5).

Finally, to assess the capability of NeuralGCM to generate tropical cyclones in an annual model integration, we use the tropical cyclone tracker TempestExtremes[33], as described in Supplementary Information section I.2, Supplementary Fig. 34 and Supplementary Table 6. Figure 4e–g shows that NeuralGCM, even at a coarse resolution of 1.4°, produces realistic trajectories and counts of tropical cyclone (83 versus 86 in ERA5 for the corresponding period), whereas X-SHiELD, when regridded to 1.4° resolution, substantially underestimates the tropical cyclone count (40). Additional statistical analyses of tropical cyclones can be found in Extended Data Figs. 7 and 8.

### Decadal simulations

To assess the capability of NeuralGCM to simulate historical temperature trends, we conduct AMIP-like simulations over a duration of 40 years with NeuralGCM-2.8°. Out of 37 different runs with initial conditions spaced every 10 days during the year 1980, 22 simulations were stable for the entire 40-year period, and our analysis focuses on these results. We compare with 22 simulations run with prescribed SST from the Coupled Model Intercomparison Project Phase 6 (CMIP6)[34], listed in Supplementary Information section I.3.

We find that all 40-year simulations of NeuralGCM, as well as the mean of the 22 AMIP runs, accurately capture the global warming trends observed in ERA5 data (Fig. 4b). There is a strong correlation in the year-to-year temperature trends with ERA5 data, suggesting that NeuralGCM effectively captures the impact of SST forcing on climate. When comparing spatial biases averaged over 1981–2014, we find that all 22 NeuralGCM-2.8° runs have smaller bias than the CMIP6 AMIP runs, and this result remains even when removing the global temperature bias in CMIP6 AMIP runs (Fig. 4c and Supplementary Figs. 32 and 33).

Next, we investigated the vertical structure of tropical warming trends, which climate models tend to overestimate in the upper troposphere[35]. As shown in Fig. 4d, the trends, calculated by linear regression, of NeuralGCM are closer to ERA5 than those of AMIP runs. In particular, the bias in the upper troposphere is reduced. However, NeuralGCM does show a wider spread in its predictions than the AMIP runs, even at levels near the surface where temperatures are typically more constrained by prescribed SST.

Lastly, we evaluated NeuralGCM's capability to generalize to unseen warmer climates by conducting AMIP simulations with increased SST (Supplementary Information section I.4.2). We find that NeuralGCM shows some of the robust features of climate warming response to modest SST increases (+1 K and +2 K); however, for more substantial SST increases (+4 K), NeuralGCM's response diverges from expectations (Supplementary Fig. 37). In addition, AMIP simulations with increased SST show climate drift, underscoring NeuralGCM's limitations in this context (Supplementary Fig. 38).

## Discussion

NeuralGCM is a differentiable hybrid atmospheric model that combines the strengths of traditional GCMs with machine learning for weather forecasting and climate simulation. To our knowledge, NeuralGCM is the first machine-learning-based model to make accurate ensemble weather forecasts, with better CRPS than state-of-the-art physics-based models. It is also, to our knowledge, the first hybrid model that achieves comparable spatial bias to global cloud-resolving models, can simulate realistic tropical cyclone tracks and can run AMIP-like simulations with realistic historical temperature trends. Overall, NeuralGCM demonstrates that incorporating machine learning is a viable alternative to building increasingly detailed physical models[32] for improving GCMs.

Compared with traditional GCMs with similar skill, NeuralGCM is computationally efficient and low complexity. NeuralGCM runs at 8- to 40-times-coarser horizontal resolution than ECMWF's Integrated Forecasting System and global cloud-resolving models, which enables 3 to 5 orders of magnitude savings in computational resources. For example, NeuralGCM-1.4° simulates 70,000 simulation days in 24 hours using a single tensor-processing-unit versus 19 simulated days on 13,824 central-processing-unit cores with X-SHiELD (Extended Data Table 1). This can be leveraged for previously impractical tasks such as large ensemble forecasting. NeuralGCM's dynamical core uses global spectral methods[36], and learned physics is parameterized with fully connected neural networks acting on single vertical columns. Substantial headroom exists to pursue higher accuracy using advanced numerical methods and machine-learning architectures.

Our results provide strong evidence for the disputed hypothesis[37–39] that learning to predict short-term weather is an effective way to tune parameterizations for climate. NeuralGCM models trained on 72-hour forecasts are capable of realistic multi-year simulation. When provided with historical SSTs, they capture essential atmospheric dynamics such as seasonal circulation, monsoons and tropical cyclones. However, we will probably need alternative training strategies[38,39] to learn important processes for climate with subtle impacts on weather timescales, such as a cloud feedback.

The NeuralGCM approach is compatible with incorporating either more physics or more machine learning, as required for operational weather forecasts and climate simulations. For weather forecasting, we expect that end-to-end learning[40] with observational data will allow for better and more relevant predictions, including key variables such as precipitation. Such models could include neural networks acting as corrections to traditional data assimilation and model diagnostics. For climate projection, NeuralGCM will need to be reformulated to enable coupling with other Earth-system components (for example, ocean and land), and integrating data on the atmospheric chemical composition (for example, greenhouse gases and aerosols). There are also research challenges common to current machine-learning-based climate models[19], including the capability to simulate unprecedented climates (that is, generalization), adhering to physical constraints, and resolving numerical instabilities and climate drift. NeuralGCM's flexibility to incorporate physics-based models (for example, radiation) offers a promising avenue to address these challenges.

Models based on physical laws and empirical relationships are ubiquitous in science. We believe the differentiable hybrid modelling approach of NeuralGCM has the potential to transform simulation for a wide range of applications, such as materials discovery, protein folding and multiphysics engineering design.

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

# Methods

## Differentiable atmospheric model

NeuralGCM combines components of the numerical solver and flexible neural network parameterizations. Simulation in time is carried out in a coordinate system suitable for solving the dynamical equations of the atmosphere, describing large-scale fluid motion and thermodynamics under the influence of gravity and the Coriolis force.

Our differentiable dynamical core is implemented in JAX, a library for high-performance code in Python that supports automatic differentiation[42]. The dynamical core solves the hydrostatic primitive equations with moisture, using a horizontal pseudo-spectral discretization and vertical sigma coordinates[36,43]. We evolve seven prognostic variables: vorticity and divergence of horizontal wind, temperature, surface pressure, and three water species (specific humidity, and specific ice and liquid cloud water content).

Our learned physics module uses the single-column approach of GCMs[2], whereby information from only a single atmospheric column is used to predict the impact of unresolved processes occurring within that column. These effects are predicted using a fully connected neural network with residual connections, with weights shared across all atmospheric columns (Supplementary Information section C.4).

The inputs to the neural network include the prognostic variables in the atmospheric column, total incident solar radiation, sea-ice concentration and SST (Supplementary Information section C.1). We also provide horizontal gradients of the prognostic variables, which we found improves performance[44]. All inputs are standardized to have zero mean and unit variance using statistics precomputed during model initialization. The outputs are the prognostic variable tendencies scaled by the fixed unconditional standard deviation of the target field (Supplementary Information section C.5).

To interface between ERA5[14] data stored in pressure coordinates and the sigma coordinate system of our dynamical core, we introduce encoder and decoder components (Supplementary Information section D). These components perform linear interpolation between pressure levels and sigma coordinate levels. We additionally introduce learned corrections to both encoder and decoder steps (Supplementary Figs. 4–6), using the same column-based neural network architecture as the learned physics module. Importantly, the encoder enables us to eliminate the gravity waves from initialization shock[45], which otherwise contaminate forecasts.

Figure 1a shows the sequence of steps that NeuralGCM takes to make a forecast. First, it encodes ERA5 data at $t = t_0$ on pressure levels to initial conditions on sigma coordinates. To perform a time step, the dynamical core and learned physics (Fig. 1b) then compute tendencies, which are integrated in time using an implicit–explicit ordinary differential equation solver[46] (Supplementary Information section E and Supplementary Table 2). This is repeated to advance the model from $t = t_0$ to $t = t_{final}$. Finally, the decoder converts predictions back to pressure levels.

The time-step size of the ODE solver (Supplementary Table 3) is limited by the Courant–Friedrichs–Lewy condition on dynamics, and can be small relative to the timescale of atmospheric change. Evaluating learned physics is approximately 1.5 times as expensive as a time step of the dynamical core. Accordingly, following the typical practice for GCMs, we hold learned physics tendencies constant for multiple ODE time steps to reduce computational expense, typically corresponding to 30 minutes of simulation time.

## Deterministic and stochastic models

We train deterministic NeuralGCM models using a combination of three loss functions (Supplementary Information section G.4) to encourage accuracy and sharpness while penalizing bias. During the main training phase, all losses are defined in a spherical harmonics basis. We use a standard mean squared error loss for prompting accuracy, modified to progressively filter out contributions from higher total wavenumbers at longer lead times (Supplementary Fig. 8). This filtering approach tackles the 'double penalty problem'[47] as it prevents the model from being penalized for predicting high-wavenumber features in incorrect locations at later times, especially beyond the predictability horizon. A second loss term encourages the spectrum to match the training data using squared loss on the total wavenumber spectrum of prognostic variables. These first two losses are evaluated on both sigma and pressure levels. Finally, a third loss term discourages bias by adding mean squared error on the batch-averaged mean amplitude of each spherical harmonic coefficient. For analysis of the impact that various loss functions have, refer to Supplementary Information section H.6.1, and Supplementary Figs. 23 and 24. The combined action of the three training losses allow the resulting models trained on 3-day rollouts to remain stable during years-to-decades-long climate simulations. Before final evaluations, we perform additional fine-tuning of just the decoder component on short rollouts of 24 hours (Supplementary Information section G.5).

Stochastic NeuralGCM models incorporate inherent randomness in the form of additional random fields passed as inputs to neural network components. Our stochastic loss is based on the CRPS[28,48,49]. CRPS consists of mean absolute error that encourages accuracy, balanced by a similar term that encourages ensemble spread. For each variable we use a sum of CRPS in grid space and CRPS in the spherical harmonic basis below a maximum cut-off wavenumber (Supplementary Information section G.6). We compute CRPS on rollout lengths from 6 hours to 5 days. As illustrated in Fig. 1, we inject noise to the learned encoder and the learned physics module by sampling from Gaussian random fields with learned spatial and temporal correlation (Supplementary Information section C.2 and Supplementary Fig. 2). For training, we generate two ensemble members per forecast, which suffices for an unbiased estimate of CRPS.

## Data availability

For training and evaluating the NeuralGCM models, we used the publicly available ERA5 dataset[14], originally downloaded from https://cds.climate.copernicus.eu/ and available via Google Cloud Storage in Zarr format at gs://gcp-public-data-arco-era5/ar/full_37-1h-0p25deg-chunk-1.zarr-v3. To compare NeuralGCM with operational and data-driven weather models, we used forecast datasets distributed as part of WeatherBench2[12] at https://weatherbench2.readthedocs.io/en/latest/data-guide.html, to which we have added NeuralGCM forecasts for 2020. To compare NeuralGCM with atmospheric models in climate settings, we used CMIP6 data available at https://catalog.pangeo.io/browse/master/climate/, as well as X-SHiELD[24] outputs available on Google Cloud storage in a 'requester pays' bucket at gs://ai2cm-public-requester-pays/C3072-to-C384-res-diagnostics. The Radiosonde Observation Correction using Reanalyses (RAOBCORE) V1.9 that was used as reference tropical temperature trends was downloaded from https://webdata.wolke.img.univie.ac.at/haimberger/v1.9/. Base maps use freely available data from https://www.naturalearthdata.com/downloads/.

## Code availability

The NeuralGCM code base is separated into two open source projects: Dinosaur and NeuralGCM, both publicly available on GitHub at https://github.com/google-research/dinosaur (ref. 50) and https://github.com/google-research/neuralgcm (ref. 51). The Dinosaur package implements a differentiable dynamical core used by NeuralGCM, whereas the NeuralGCM package provides machine-learning models and checkpoints of trained models. Evaluation code for NeuralGCM weather forecasts is included in WeatherBench2[12], available at https://github.com/google-research/weatherbench2 (ref. 52).

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

**Acknowledgements** We thank A. Kwa, A. Merose and K. Shah for assistance with data acquisition and handling; L. Zepeda-Núñez for feedback on the paper; and J. Anderson, C. Van Arsdale, R. Chemke, G. Dresdner, J. Gilmer, J. Hickey, N. Lutsko, G. Nearing, A. Paszke, J. Platt, S. Ponda, M. Pritchard, D. Rothenberg, F. Sha, T. Schneider and O. Voicu for discussions.

**Author contributions** D.K., J.Y., I.L., P.N., J.S. and S. Hoyer contributed equally to this work. D.K., J.Y., I.L., P.N., J.S., G.M., J.L. and S. Hoyer wrote the code. D.K., J.Y., I.L., P.N., G.M. and S. Hoyer trained models and analysed the data. M.P.B. and S. Hoyer managed and oversaw the research project. M.K., S.R., P.D., S. Hatfield, P.B. and M.P.B. contributed technical advice and ideas. M.W. ran experiments with GraphCast for comparison with NeuralGCM. A.S.-G. assisted with data preparation. D.K., J.Y., I.L., P.N. and S. Hoyer wrote the paper. All authors gave feedback and contributed to editing the paper.

**Competing interests** D.K., J.Y., I.L., P.N., J.S., J.L., S.R., P.B., A.S.-G., M.W., M.P.B. and S. Hoyer are employees of Google. S. Hoyer, D.K., I.L., J.Y., G.M., P.N., J.S. and M.B. have filed international patent application PCT/US2023/035420 in the name of Google LLC, currently pending, relating to neural general circulation models.

**Additional information**
**Correspondence and requests for materials** should be addressed to Dmitrii Kochkov, Janni Yuval or Stephan Hoyer.

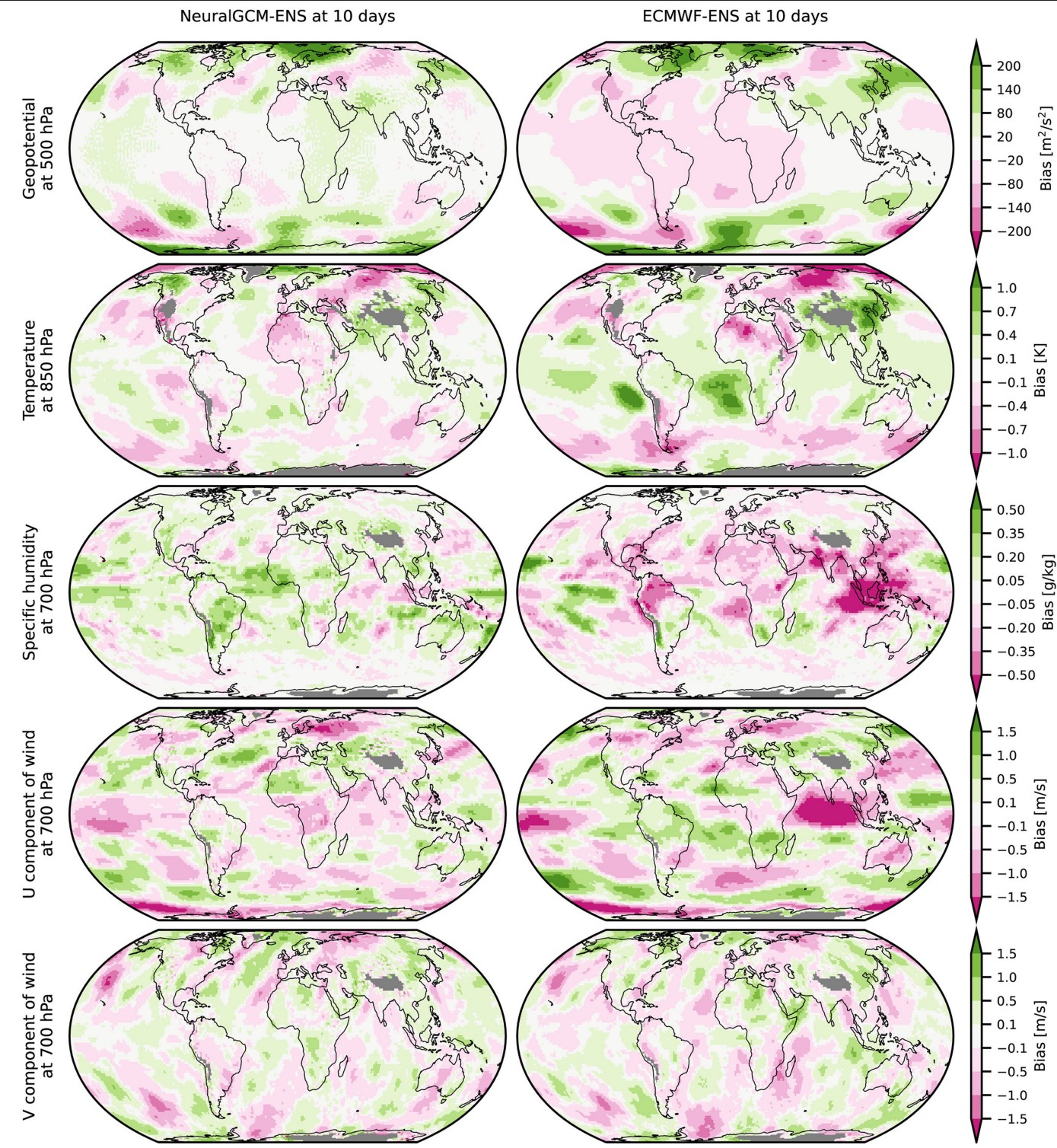

**Extended Data Fig. 1 | Maps of bias for NeuralGCM-ENS and ECMWF-ENS forecasts.** Bias is averaged over all forecasts initialized in 2020.

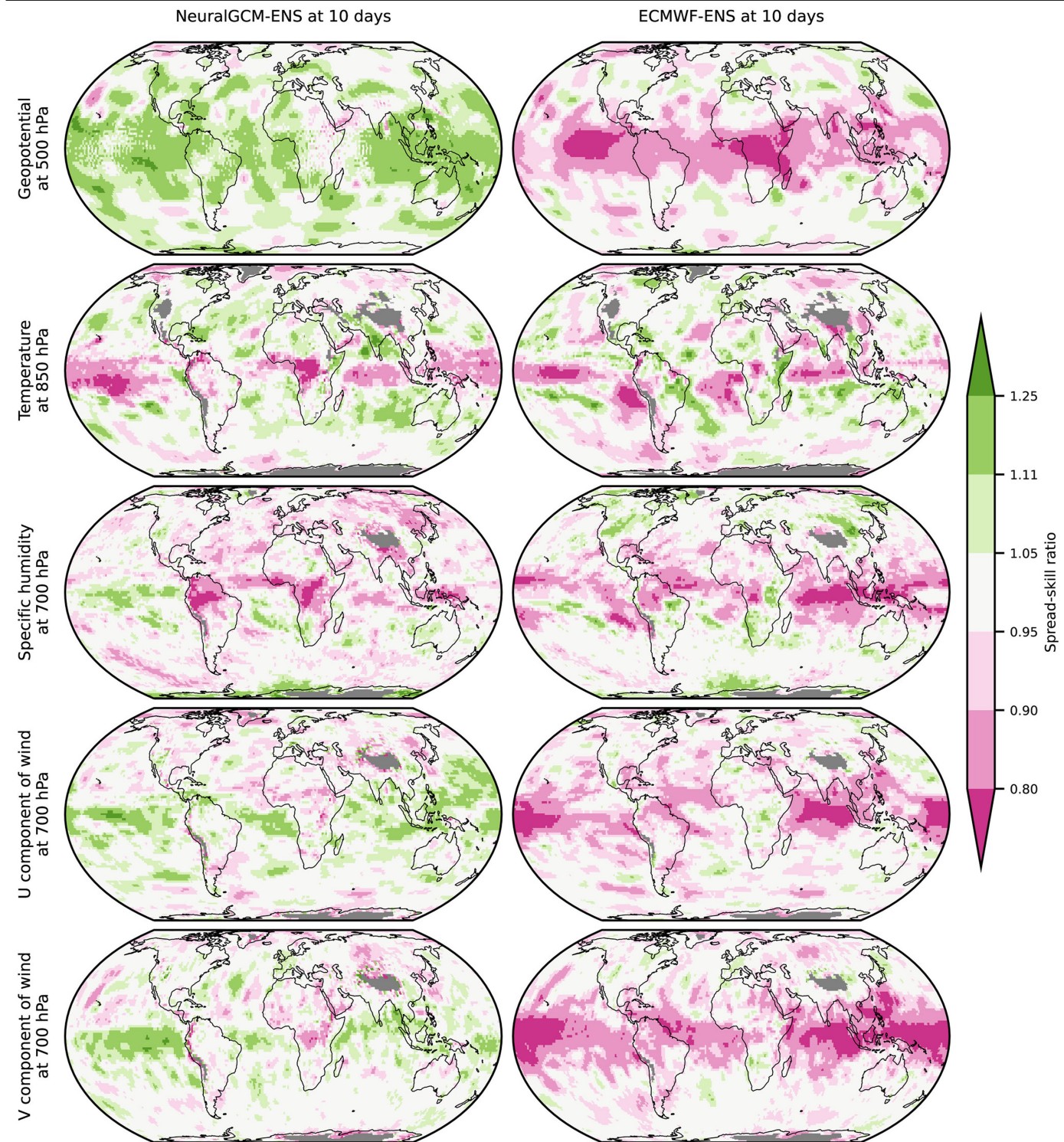

**Extended Data Fig. 2 | Maps of spread-skill ratio for NeuralGCM-ENS and ECMWF-ENS forecasts.** Spread-skill ratio is averaged over all forecasts initialized in 2020.

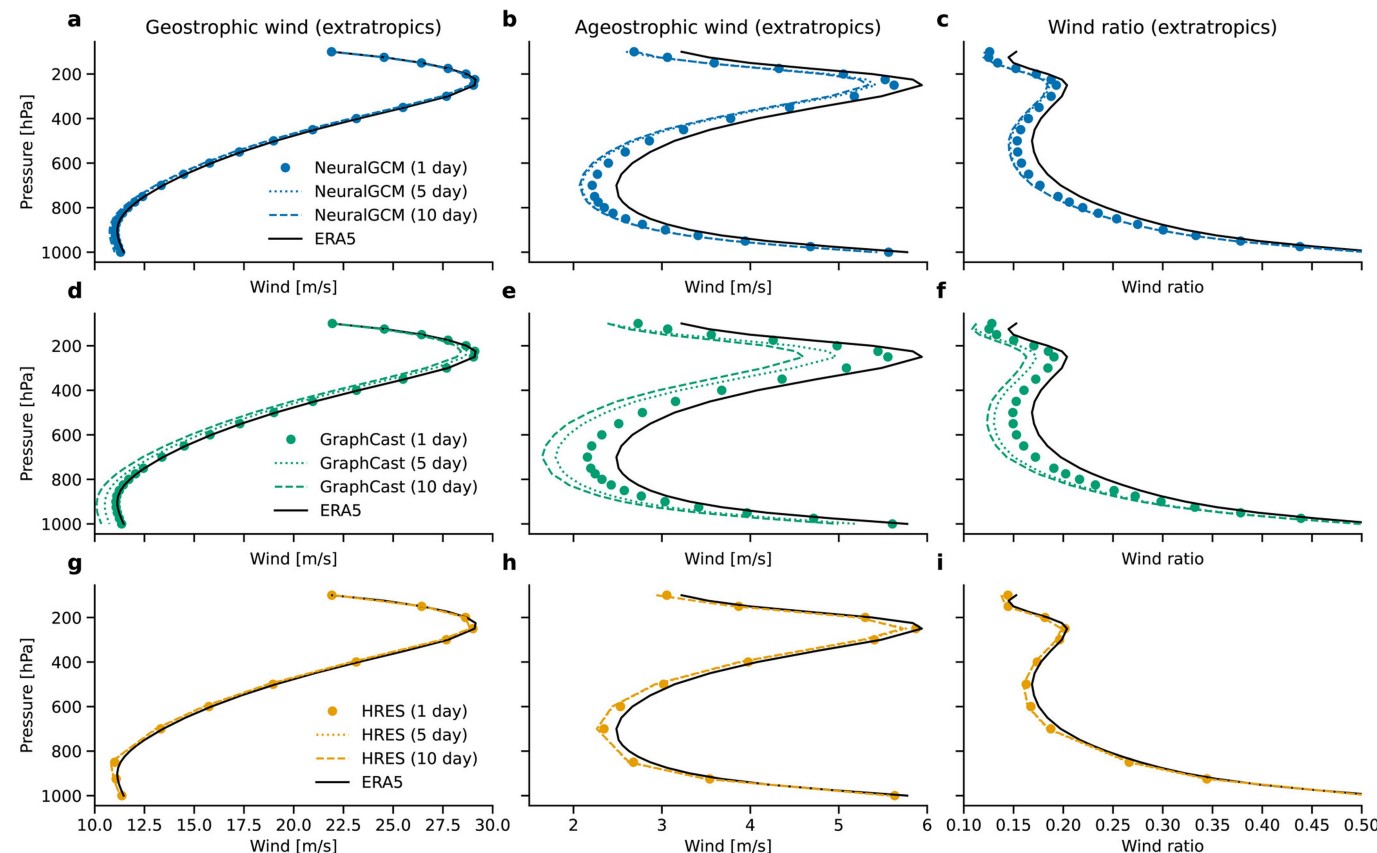

**Extended Data Fig. 3 | Geostrophic balance in NeuralGCM, GraphCast[3] and ECMWF-HRES.** Vertical profiles of the extratropical intensity (averaged between latitude 30°–70° in both hemispheres) and over all forecasts initialized in 2020 of (a,d,g) geostrophic wind, (b,e,h) ageostrophic wind and (c,f,i) the ratio of the intensity of ageostrophic wind over geostrophic wind for ERA5 (black continuous line in all panels), (a,b,c) NeuralGCM-0.7°, (d,e,f) GraphCast and (g,h,i) ECMWF-HRES at lead times of 1 day, 5 days and 10 days.

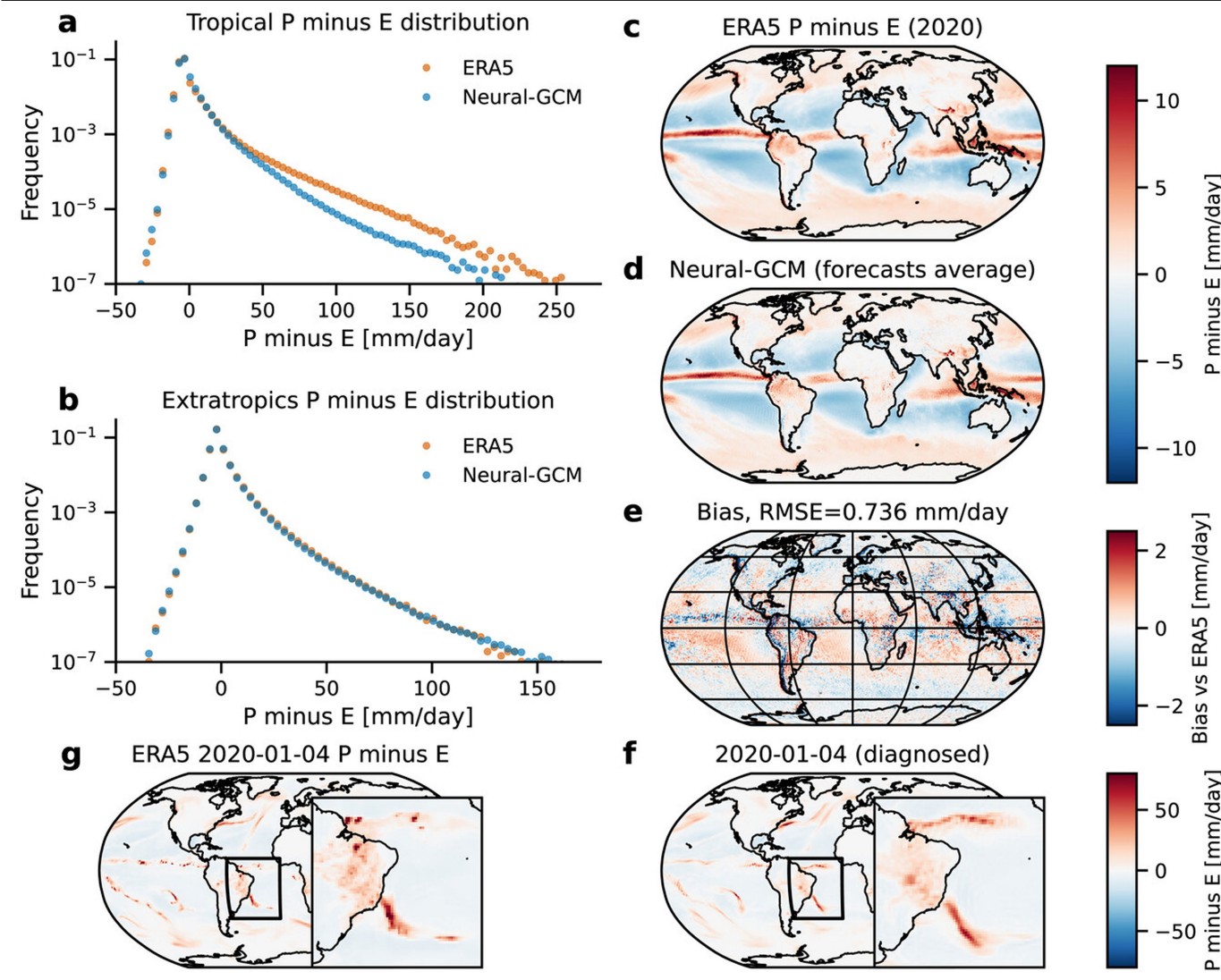

**Extended Data Fig. 4 | Precipitation minus evaporation calculated from the third day of weather forecasts.** (a) Tropical (latitudes −20° to 20°) precipitation minus evaporation (P minus E) rate distribution, (b) Extratropical (latitudes 30° to 70° in both hemispheres) P minus E, (c) mean P minus E for 2020 ERA5[14] and (d) NeuralGCM-0.7° (calculated from the third day of forecasts and averaged over all forecasts initialized in 2020), (e) the bias between NeuralGCM-0.7° and ERA5, (f-g) Snapshot of daily precipitation minus evaporation for 2020-01-04 for (f) NeuralGCM-0.7° (forecast initialized on 2020-01-02) and (g) ERA5.

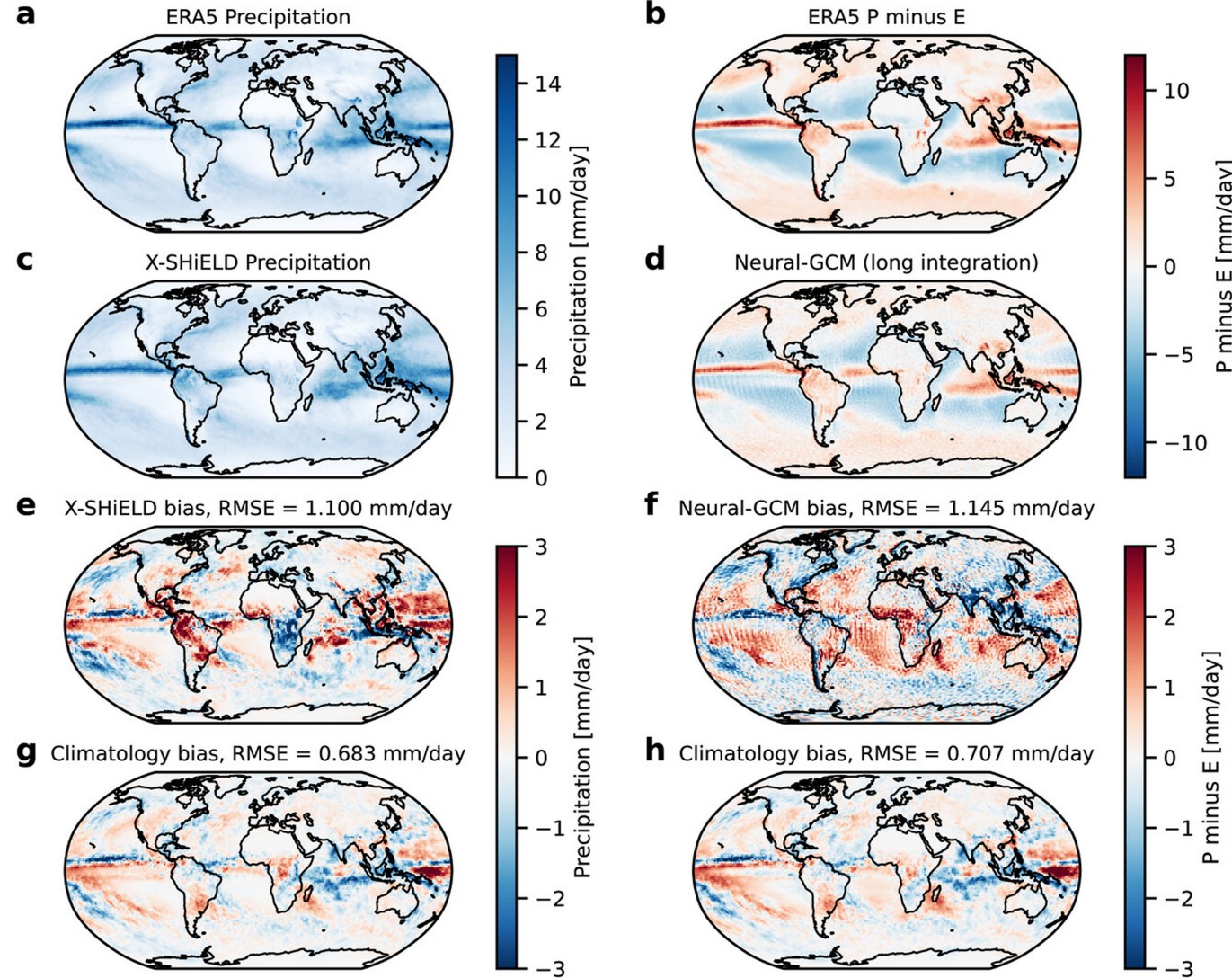

**Extended Data Fig. 5 | Indirect comparison between precipitation bias in X-SHiELD and precipitation minus evaporation bias in NeuralGCM-1.4°.** Mean precipitation calculated between 2020-01-19 and 2021-01-17 for (a) ERA5[14] (c) X-SHiELD[31] and the biases in (e) X-SHiELD and (g) climatology (ERA5 data averaged over 1990-2019). Mean precipitation minus evaporation calculated between 2020-01-19 and 2021-01-17 for (b) ERA5 (d) NeuralGCM-1.4° (initialized in October 18th 2019) and the biases in (f) NeuralGCM-1.4° and (h) climatology (data averaged over 1990–2019).

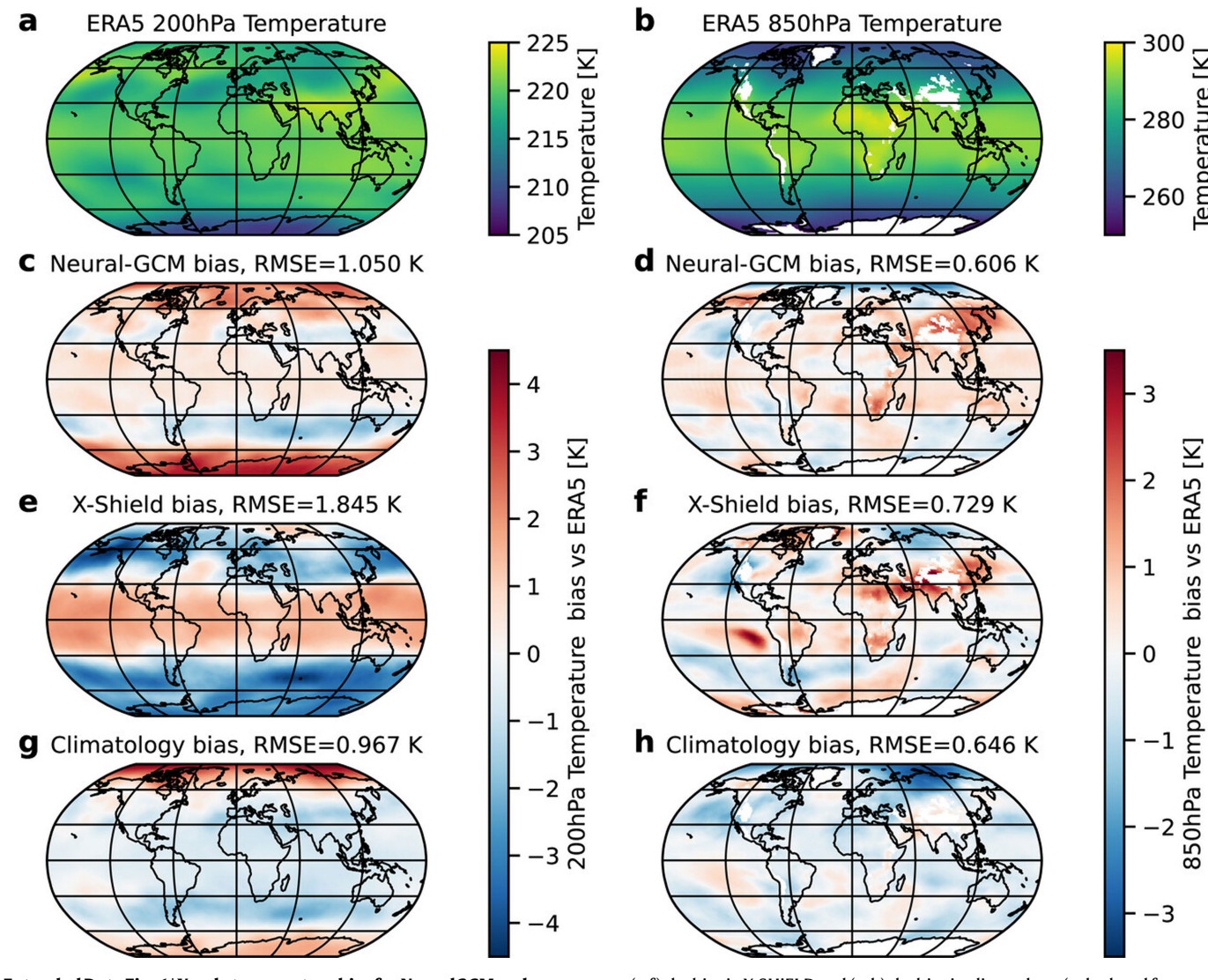

**Extended Data Fig. 6 | Yearly temperature bias for NeuralGCM and X-SHiELD[31].** Mean temperature between 2020-01-19 to 2020-01-17 for (a) ERA5 at 200hPa and (b) 850hPa. (c,d) the bias in the temperature for NeuralGCM-1.4°, (e,f) the bias in X-SHiELD and (g,h) the bias in climatology (calculated from 1990–2019). NeuralGCM-1.4° was initialized in 18th of October (similar to X-SHiELD).

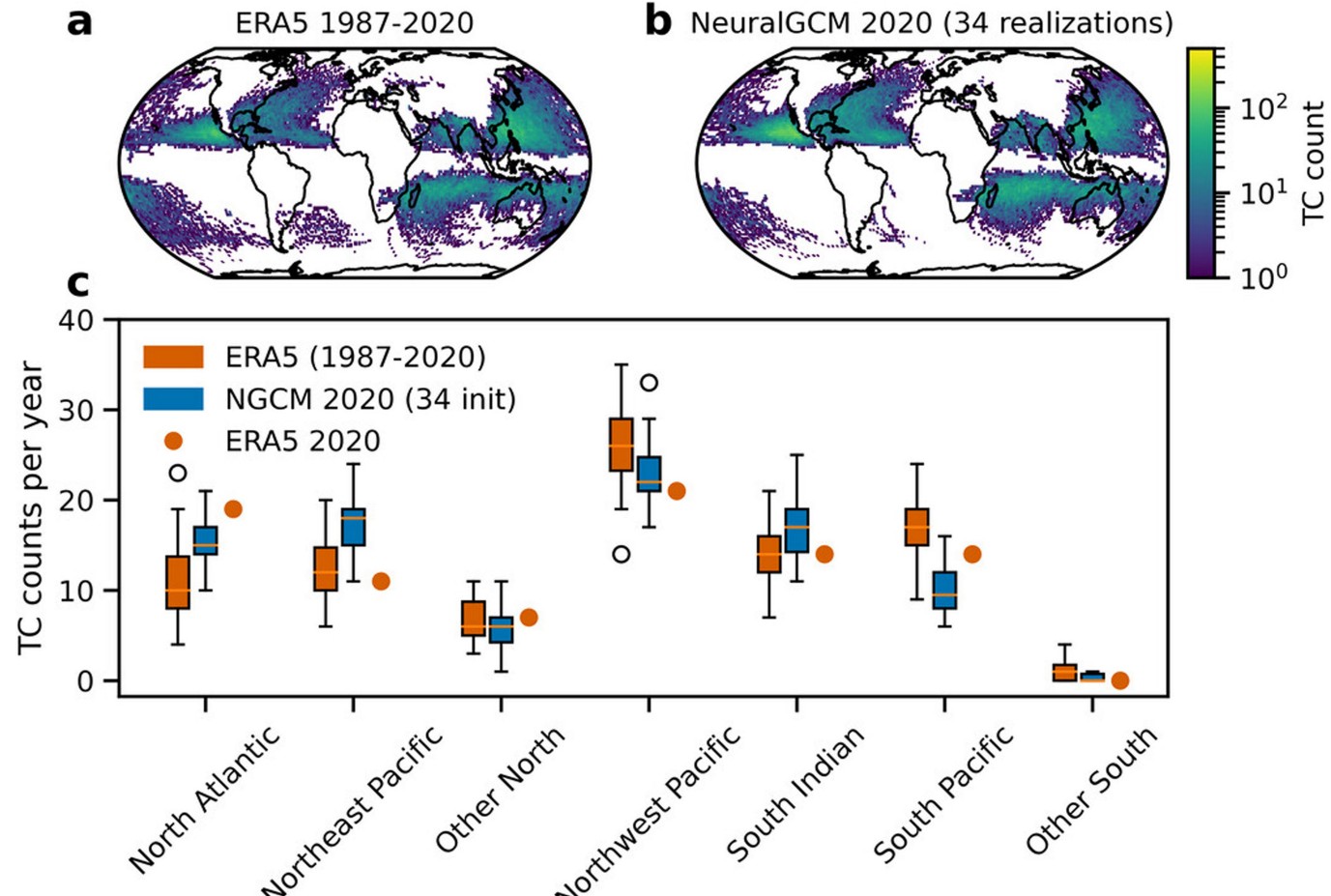

**Extended Data Fig. 7 | Tropical Cyclone densities and annual regional counts.** (a) Tropical Cyclone (TC) density from ERA5[14] data spanning 1987–2020. (b) TC density from NeuralGCM-1.4° for 2020, generated using 34 different initial conditions all initialized in 2019. (c) Box plot depicting the annual number of TCs across different regions, based on ERA5 data (1987–2020), NeuralGCM-1.4° for 2020 (34 initial conditions), and orange markers show ERA5 for 2020. In the box plots, the red line represents the median; the box delineates the first to third quartiles; the whiskers extend to 1.5 times the interquartile range (Q1 − 1.5IQR and Q3 + 1.5IQR), and outliers are shown as individual dots. Each year is defined from January 19th to January 17th of the following year, aligning with data availability from X-SHiELD. For NeuralGCM simulations, the 3 initial conditions starting in January 2019 exclude data for January 17th, 2021, as these runs spanned only two years.

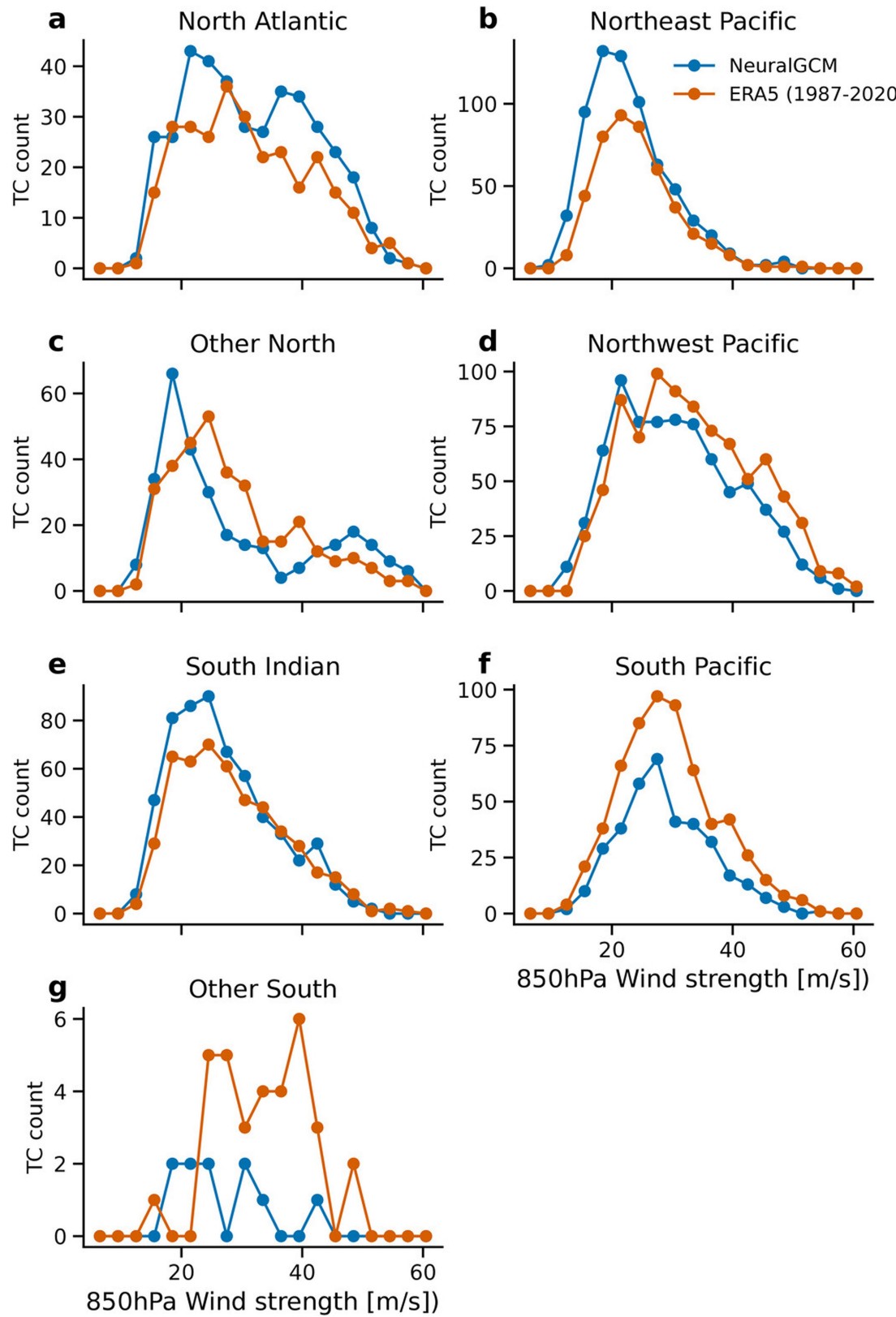

**Extended Data Fig. 8 | Tropical Cyclone maximum wind distribution in NeuralGCM vs. ERA5**[14]**.** Number of Tropical Cyclones (TCs) as a function of maximum wind speed at 850hPa across different regions, based on ERA5 data (1987–2020; in orange), and NeuralGCM-1.4° for 2020 (34 initial conditions; in blue). Each year is defined from January 19th to January 17th of the following year, aligning with data availability from X-SHiELD. For NeuralGCM simulations, the 3 initial conditions starting in January 2019 exclude data for January 17th, 2021, as these runs spanned only two years.

**Extended Data Table 1 | Resource requirements for different NeuralGCM models trained on Google Cloud TPUs**

|  | NeuralGCM-2.8° | NeuralGCM-1.4° | NeuralGCM-0.7° | NeuralGCM-ENS (1.4°) |
|---|---|---|---|---|
| Training time | 1 day | 1 week | 3 weeks | 10 days |
| Device | 16 TPU v4 | 16 TPU v4 | 256 TPU v4 | 128 TPU v5e |
| Parallelism | batch=16 | batch=16 | batch=16<br>x=2<br>y=2<br>z=4 | batch=16<br>x=2<br>z=2<br>ensemble=2 |
| Parameter count | 14.5 M | 18.3 M | 31.1 M | 11.5 M |
| Inference time (10 day forecast) | 2.5s | 12.6s | 119s | 12.4s |
| Inference speed (sim days/day) | 350 000 | 69 000 | 7300 | 70 000 |
| Peak memory usage during inference | 255MB | 1100MB | 4177MB | 1011MB |

Inference time and speed at reported for a single core TPU v4.