## [Peer Review File · Nature]

Manuscript Title: Neural General Circulation Models for Weather and Climate

Reviewer Comments & Author Rebuttals

Reviewer Reports on the Initial Version:

Referees' comments:

Referee #1 (Remarks to the Author):

This article proposes a novel fully differentiable hybrid physics-based numerical dynamical core with machine learned components, called NeuralGCM, and demonstrate remarkable skill and performance on a slew of ML and physical metrics across timescales (weather and climate). The deterministic and ensemble skill of NeuralGCM for both weather and climate prediction is impressive and represents a step-change in the field of hybrid physics-ML weather and climate modeling. For the following key reasons I recommend the paper for publication, after the authors have addressed my primary concerns and comments (further below):

1. NeuralGCM enables end-to-end online training of a hybrid physics-ML model that enables stable and accurate forecasts, thus solving a plaguing (instability) problem in most other hybrid approaches thus far.
2. NeuralGCM proves that column-local computations are sufficient for fairly accurate and stable predictions over long rollouts, which supports an elegant and simple formulation of the hybrid approach.
3. NeuralGCM introduces a parameter-free Gaussian random field noise ansatz, that has learned spatial and temporal correlations, but is independent of the input fields themselves (fed in as a separate feature). This sidesteps the challenging and complicated task of tuning and calibrating parameters in other approaches to modeling initial condition and model uncertainty, which has been the primary challenge for the generation of reliable well-calibrated large ensembles of ML-based models.
4. NeuralGCM achieves lower error (better RMSE, RMSB, and CRPS) compared to the gold standard for ensemble prediction, ECMWF-ENS, which in and of itself is a significant achievement. Further, NeuralGCM achieves this at 8x coarser resolution!
5. At a lower resolution than most ML-based forecasting models (0.7-degree vs. 0.25-degree), NeuralGCM outperforms other models not only on skill but also on forecast sharpness, which is critical for extreme weather events. Relatedly, NeuralGCM's spectra remain consistent over lead time, and its representation of the vertical structure of geostrophic wind does not show significant degradation beyond day 5.
6. NeuralGCM shows signs of good generalization to future climate through its roughly similar error for initialization further out into the future. However, more rigorous analyses are required to convince the reader of its generalization characteristics.
7. NeuralGCM proves high reliability on a host of climate metrics: (a) a reliable seasonal cycle of

precipitable water and total KE, Hadley circulation, spatial patterns of Eddy KE, and the monsoons; (b) surprisingly accurate tropical cyclone statistics even at coarse resolutions; (c) reduction in biases compared to the AMIP ensemble; (d) comparable biases to global CRMs.

8. NeuralGCM offers evidence that strengthens the hypothesis that training on short timescales is an effective way to tune other learned components of the Earth system for climate timescales, although much work remains to be done in understanding the mechanisms of instability in some of NeuralGCM's long rollouts.

This article clearly represents a Herculean effort with keen attention to detail and a robust set of analyses, metrics, and diagnostics to uncover the physical consistency and characteristics of the model. 50 pages of material in the appendices in and of itself shows the breadth and depth of analyses in this work! Many aspects of the work are novel and intriguing and the authors have taken care to justify and validate choices of the model design and training protocols. The article is presented in a compelling fashion, although some work could be done to restructure the paper in a way that highlights the most significant accomplishments and their implications for the future of ML-based weather and climate prediction. Presently, some of the key kernels are somewhat dispersed and drowned out in the vast amount of detail. Specifically, the introduction, discussion and conclusions could use some work to improve clarity, lucidity, and punch.

I have a few major concerns and several minor comments or questions.

Major:

1. Scalability. How does the model training cost and inference latency scale with resolution and complexity? Also, can some analysis be presented on how the computational performance of the model (memory consumption, bandwidth, IO, data movement, etc.) would scale with resolution and other parameters? How would the model perform on GPUs vs. TPUs? What do the profiles reveal about what bottlenecks might emerge in the future at higher resolution and/or greater complexity / more input/output fields, especially with training over so many steps (72!)? Importantly, how would the model scale into the non-hydrostatic regime, looking ahead towards kilometer-scale prediction? The current training cost at 0.7-degree resolution is 130,000 TPU-hours, which gives me serious concern about the scalability of this model.

2. Ablations. The article could use more ablations to convince the reader about reasons for specific and somewhat curious design choices. The most significant is the loss function. The combination of terms, although some explanations are given, appears to be a mish-mash of things that just worked. Although this is not uncommon in ML, where the end sometimes justifies the means, it would be helpful to provide ablations for high-wavenumber-filtering, bias-correction, spectral-matching, multi-step-training, spectral cutoffs, etc. Additionally, it might be helpful to describe the computational cost of these complex loss functions. Other areas that could use ablations are the roles of the encoder and decoder. It is unclear, for example, how the encoder interacts with multi-step training. Similarly, why were certain normalization and scaling choices made versus other possible strategies?

3. Interpretability. An attractive property of hybrid models is their interpretability compared to full

model emulators like GraphCast. However, it appears that a lot of the interpretability in NeuralGCM is lost by design, because of the decoder. Some commentary on how this could be avoided or done differently would be satisfying. I imagine this loss in interpretability would likely hurt the potential operational use of the model (for climate).

4. Generalization. Although the results shown are very impressive, more work remains on proving NeuralGCM's extrapolation and generalization properties. Could the model be tested on plausible future SST scenarios (within the AMIP framework). Could greenhouse gas concentrations be incorporated into the model? What about other climate change markers? Could there be a climate-invariant reformulation of NeuralGCM and its inputs? Would NeuralGCM benefit from training on future climate simulations?

5. Instability. It is somewhat worrisome that some members of NeuralGCM blow up. One might expect that with a numerical dycore the model should be stable, especially in the hydrostatic limit. But this is not the case and it is unclear why.

Minor:

1. Can some parallels be drawn to the rapidly exploding field of diffusion modeling?
2. Is multi-step training needed? Can the model be trained on fewer steps, shorter timescales?
3. Can the loss function be posed more elegantly and simply?
4. Why are two ensemble members sufficient for the stochastic training? Is there a mathematical explanation?
5. Can the numerical dycore be done away with? Perhaps using a lightweight ML surrogate?
6. Isn't the spectral loss the same as the L2 loss in physical space (from generalized Parseval's theorem)?
7. How does time-step filtering and the other loss functions affect the effective resolution of the model?
8. Can NeuralGCM be reformulate to predict both precipitation and evaporation?
9. Why does the model under-predict extremes in the tropics?
10. Why does the model have grid-scale artifacts (Fig. I-26)?
11. Why does the model have a larger spread than AMIP runs, even close to the surface where temperatures are expected to be more constrained by SST?

Referee #2 (Remarks to the Author):

The paper presents a novel, fully differentiable hybrid ML-physics general circulation model. This is indeed impressive work. It addresses the instability of long rollouts in pure-AI weather prediction models by incorporating explicit large-scale physics. Additionally, it overcomes the limitations of traditional hybrid ML-physics climate modeling, critically hindered by non-differentiable Fortran codebases. The formulation of the domain knowledge-infused loss function is particularly creative and effective. The manuscript is generally clear, thoroughly incorporating supplementary materials. However, there are several areas where clarity could be enhanced:

1. The novelty and contribution of this work seem understated. The authors should more clearly

articulate the work's impact and contribution in the introduction.

2. The paper does not adequately address the limitations of this work. It gives the impression that this model solves all climate modeling problems by breaking the trade-offs between computational cost and accuracy. A more detailed discussion of limitations, perhaps referring to the 'SILIENT' principle in Bretherton (2023), would be beneficial.

3. I suggest the authors provide recommendations for future research, outlining both immediate and long-term steps for those inspired by this work.

4. A more detailed visual presentation of the NGCM is needed (maybe in the SI?). While Figure 1 shows the overall structure, it lacks details on components like the encoder, decoder, and learned physics (spatial embeddings and block architecture).

5. A complete description of input and output variables is missing. A table summarizing all input, output, and forcing variables would be helpful. This is especially helpful since the forcing variables are usually treated as input variables in similar works.

6. The comparison of NGCM with XSHIELDS and AMIP models (L443-454 and L502-514) is a bit one-sided. These models are built on general atmospheric physics principles, unlike NGCM, which is explicitly trained to emulate ERA5. The nuanced differences, especially the lesser tuning of XSHIELDS due to computational costs, should be highlighted.

7. Regarding the stability of hybrid climate modeling (L498-500), including common failure modes in the SI would be beneficial for hybrid model developers.

8. Information about the model code repository and its accessibility is missing. Is it going to be open-sourced?

9. Several important references are missing and should be incorporated:

a. Pathak, Jaideep, et al. "Fourcastnet: A global data-driven high-resolution weather model using adaptive fourier neural operators." arXiv preprint arXiv:2202.11214 (2022).

b. Nguyen, Tung, et al. "ClimaX: A foundation model for weather and climate." arXiv preprint arXiv:2301.10343 (2023).

c. Watt-Meyer, Oliver, et al. "ACE: A fast, skillful learned global atmospheric model for climate prediction." arXiv preprint arXiv:2310.02074 (2023).

d. Han, Yilun, Guang J. Zhang, and Yong Wang. "An ensemble of neural networks for moist physics processes, its generalizability and stable integration." *Journal of Advances in Modeling Earth Systems* 15.10 (2023): e2022MS003508.

e. Hakim, Gregory J., and Sanjit Masanam. "Dynamical Tests of a Deep-Learning Weather Prediction Model." arXiv preprint arXiv:2309.10867 (2023).

10. The order of figure references is not consistent (e.g., L437-442). This should be corrected throughout the manuscript.

11. The title is too vague, not efficiently conveying the topic of the manuscript.

12. Minor comments:

a. More description is needed for regriding methods beyond "conservative."

b. L189 vs. L208: There's inconsistency in the rollout schedule

c. L212-214: Clarification on how randomness is incorporated

d. L263-264: Generalizability compared to other models and variables/levels

e. L271: What about 0.7deg version?

f. Fig 2: Clarification of grey areas is missing

g. L1017-2020: How were the land surfaces treated?

h. Fig H3: What makes GraphCast more accurate for a short lead time? Or, What makes NGCM less accurate for a short lead time?

i. Figs H7 and H8: Is there any explanation for common behaviors (e.g., dipping and recovering CRPS around 3-5 day's mark; the initial increase of the spread-skill ratio)?

Referee #3 (Remarks to the Author):

A. Summary of the key results

This study presents the first general circulation model which successfully uses a neural network for all subgrid-scale "physics" to enable long accurate simulations. The key methodological advance over previous works is the use of a differentiable dynamical core which allows optimization accounting for the combined effect of dynamics and the learned physics over multiple time steps. The resulting system is able to make very accurate weather forecasts when initialized from ERA5 reanalysis and also has attractive ensemble prediction capabilities. Impressively, it is also capable of 30-year simulations with low climate biases, although about one third of simulations are unstable when run for this length.

B. Originality and significance

The successful use of a differentiable dynamical core, and the combined results of excellent weather forecast skill and mostly small climate biases are certainly novel and significant. However some prior works should be acknowledged. For example, when discussing previous hybrid dynamics + ML physics approaches, the authors do not cite a line of work that has successfully emulated cloud resolving models embedded in a GCM with realistic topography. E.g. see <https://doi.org/10.1029/2020ms002076> and <https://doi.org/10.1029/2022ms003508>.

Furthermore, the authors claim that pure ML models trained to predict weather are not capable of long-term stable simulations (lines 35-37 and 83-84). While it is true that the existing ML models which beat ECMWF's IFS model do not show forecasts beyond 14 days, promising results for year long forecasts were shown by <https://doi.org/10.1029/2020MS002109> and <https://arxiv.org/abs/2306.03838>.

Additionally, stable and accurate 10-100 year forecasts from a pure ML model were shown by <https://doi.org/10.48550/arXiv.2310.02074> (albeit trained on climate model output, instead of ERA5). These prior results should be acknowledged.

C. Data & methodology

In general the approach is appropriate given the goals of the study. However, it should be made clearer what the advantage of the hybrid dynamics + ML approach is over a pure ML strategy. For example, is the improved performance over prior approaches (say, in terms of 5-15 weather forecast skill or sharpness of longer forecasts) due to the hybrid modeling approach or is it due to the various modifications of the loss function, which could easily be applied to a pure ML model?

In my opinion, the advantages of the hybrid approach are interpretability, and possibly improved generalizability and a reduction in the amount of training data required. However these strengths are only tangentially discussed which is unfortunate.

In terms of the interpretability question, it would be useful to show some of the internal workings of NeuralGCM. For example, do the “dynamics” and “physics” tendencies in a single time step (Fig. 1) correspond to what we would expect in a typical GCM? In particular, it seems highly likely that the learned part of the tendency (which is called “physics”) actually represents some combination of subgrid-scale processes and a correction for errors introduced by the explicit dynamics. This interpretation is supported by occurrence of grid-scale noise in the P-E plots shown in Figures H21d and I26d, and it should be discussed in the main text.

D. Appropriate use of statistics and treatment of uncertainties

Yes. The analysis of ensemble forecast skill is nice to see.

E. Conclusions: robustness, validity, reliability

Many aspects of methodology (e.g. loss function adjustments which improve 5-15 day skill and sharpness of forecasts) could easily be applied to a pure-ML forecasting system. This should be acknowledged, and the particular advantages of the hybrid physics-ML system should be clarified. Additionally, it is implied that this approach could be useful for climate change simulations. How can we trust it for future or past “out of training distribution” conditions when it is only trained on historical reanalysis data?

F. Suggested improvements: experiments, data for possible revision

More timing information for inference speed should be provided (currently only the 1.4° timing is mentioned). Showing more snapshots of the “physics” tendencies would help with understanding the characteristics of the model. A stronger generalization test would be useful for showing the strengths of the hybrid ML approach. E.g. try applying a +2K or +4K sea surface temperature perturbation.

G. References: appropriate credit to previous work?

Some additional references that should be cited were listed in point 2 above.

H. Clarity and context

The writing and presentation is very clear. The appendix material is somewhat overwhelming in quantity, but useful for understanding the various strengths/weaknesses of NeuralGCM.

Below I provide a series of comments noted while reading the manuscript.

Line 36-38 and 83-84: previously published ML-based models do show promising results for long-term prediction (e.g. Weyn et al. 2020, Bonev et al. 2023, Watt-Meyer et al. 2023). Although none of those models are also able to have state-of-the-art weather forecast skill, it is not clear that this is necessary for them to be useful for climate prediction. These prior works should be acknowledged.

Line 120-122: the SPCAM emulator work has been fairly successful in realistic geography setting.

Lines 136-142: although it is called a “physics” module, if the goal is to replicate ERA5, the ML-learned component will also have to correct errors from the relatively coarse dynamical core. This needs to be stated up-front. (See also lines 986-987)

Lines 156-157: the fact that horizontal gradients help prediction is a hint that some part of the “physics module” target is a dynamics correction. The grid-scale noise in Figures H21d and I26d also indicates the same thing.

Lines 161-163: encoder-decoder is cool!

Lines 179-181: usually this is just done for radiation, not all physics.

Lines 395-400: this is a somewhat weak generalization test (although admittedly one that GraphCast does not pass). What happens if you do something like apply a +2K SST perturbation to 2020?

Line 423-424 and lines 498-499: what is the nature of the instabilities that sometimes occur?

Figure 4b: Is this also 850hPa temperature? It seems there is more ensemble spread in NeuralGCM early in the simulation—is this true, and if so do you have any idea why?

Lines 531-532: it would be useful to report the timings for other resolutions also.

Lines 553-555: do you have any ideas for ways to resolve these generalization challenges? This is absolutely critical if NeuralGCM is to be used as part of a climate model, as the manuscript seems to propose. At least suggesting a path forward here would be useful.

Line 994: what exactly do you mean by “temperature variation”?

Line 1011: “receives an additional” -> “receives additional”

Line 1089-1093: why the different configuration for different resolutions w.r.t the land/sea-ice embeddings?

Section C.4: I don’t understand how the vertical/surface embedding networks interface with the EPD architecture discussed on lines 1066-1074. A schematic would be helpful.

Section D.2: how large are the corrections to linear interpolation from the decoder NN?

Line 1566: reference not defined

Section G.7: how many trainable parameters are there for each resolution?

Figure H13: Is “NeuralGCM-HRES” the same as “NeuralGCM-0.7°”? Didn’t see the “HRES” label used elsewhere.

Figure H17: where does the negative cloud liquid/ice come from? Dynamics or the ML? It is interesting how much NeuralGCM struggles to get the zero cloud regime correct.

Figure I24a: Interesting (and likely unrealistic) big drops in precipitable water for at least one ensemble members. Any idea of what the mechanism is?

3 Reviewer 1

This article proposes a novel fully differentiable hybrid physics-based numerical dynamical core with machine learned components, called NeuralGCM, and demonstrate remarkable skill and performance on a slew of ML and physical metrics across timescales (weather and climate). The deterministic and ensemble skill of NeuralGCM for both weather and climate prediction is impressive and represents a step-change in the field of hybrid physics-ML weather and climate modeling. For the following key reasons I recommend the paper for publication, after the authors have addressed my primary concerns and comments (further below):

1. NeuralGCM enables end-to-end online training of a hybrid physics-ML model that enables stable and accurate forecasts, thus solving a plaguing (instability) problem in most other hybrid approaches thus far.

2. NeuralGCM proves that column-local computations are sufficient for fairly accurate and stable predictions over long rollouts, which supports an elegant and simple formulation of the hybrid approach.

3. NeuralGCM introduces a parameter-free Gaussian random field noise ansatz, that has learned spatial and temporal correlations, but is independent of the input fields themselves (fed in as a separate feature). This sidesteps the challenging and complicated task of tuning and calibrating parameters in other approaches to modeling initial condition and model uncertainty, which has been the primary challenge for the generation of reliable well-calibrated large ensembles of ML-based models.

4. NeuralGCM achieves lower error (better RMSE, RMSB, and CRPS) compared to the gold standard for ensemble prediction, ECMWF-ENS, which in and of itself is a significant achievement. Further, NeuralGCM achieves this at 8x coarser resolution!

5. At a lower resolution than most ML-based forecasting models (0.7-degree vs. 0.25-degree), NeuralGCM outperforms other models not only on skill but also on forecast sharpness, which is critical for extreme weather events. Relatedly, NeuralGCM's spectra remain consistent over lead time, and its representation of the vertical structure of geostrophic wind does not show significant degradation beyond day 5.

6. NeuralGCM shows signs of good generalization to future climate through its roughly similar error for initialization further out into the future. However, more rigorous analyses are required to convince the reader of its generalization characteristics.

7. NeuralGCM proves high reliability on a host of climate metrics: (a) a reliable seasonal cycle of precipitable water and total KE, Hadley circulation, spatial patterns of Eddy KE, and the monsoons; (b) surprisingly accurate tropical cyclone statistics even at coarse resolutions; (c) reduction in biases compared to the AMIP ensemble; (d) comparable biases to global CRMs.

8. NeuralGCM offers evidence that strengthens the hypothesis that training on short timescales is an effective way to tune other learned components of the Earth system for climate timescales, although much work remains to be done in understanding the mechanisms of instability in some of NeuralGCM's long rollouts.

This article clearly represents a Herculean effort with keen attention to detail and a robust set of analyses, metrics, and diagnostics to uncover the physical consistency and characteristics of the model. 50 pages of material in the appendices in and of

itself shows the breadth and depth of analyses in this work! Many aspects of the work are novel and intriguing and the authors have taken care to justify and validate choices of the model design and training protocols. The article is presented in a compelling fashion, although some work could be done to restructure the paper in a way that highlights the most significant accomplishments and their implications for the future of ML-based weather and climate prediction. Presently, some of the key kernels are somewhat dispersed and drowned out in the vast amount of detail. Specifically, the introduction, discussion and conclusions could use some work to improve clarity, lucidity, and punch.

We thank the reviewer for the careful evaluation of the paper and very useful comments.

I have a few major concerns and several minor comments or questions.

Major:

1. **Scalability.** How does the model training cost and inference latency scale with resolution and complexity? Also, can some analysis be presented on how the computational performance of the model (memory consumption, bandwidth, IO, data movement, etc.) would scale with resolution and other parameters? How would the model perform on GPUs vs. TPUs? What do the profiles reveal about what bottlenecks might emerge in the future at higher resolution and/or greater complexity / more input/output fields, especially with training over so many steps (72!)? Importantly, how would the model scale into the non-hydrostatic regime, looking ahead towards kilometer-scale prediction? The current training cost at 0.7-degree resolution is 130,000 TPU-hours, which gives me serious concern about the scalability of this model.

Cost of inference and training. Because NeuralGCM is built around a dynamical core, doubling horizontal resolution requires halving the time step like purely physics-based models. Therefore, both training and inference costs theoretically increase by a factor of 8 (accounting only for the increased number of grid points) or 16 (for large enough grids, where the cost of spherical harmonic transformations dominates) when doubling the resolution. Additional factors make scaling more complex, including the overhead of performing computation on small tensors on the TPU and model parallelism, which we employ for training for higher resolution models. Following this comment, we have added this information to table 1 (also added to the SI), which we also show below.

Analysis of the computational performance. We have now added a summary of inference times and peak memory usage for all of the NeuralGCM models (table 1 which was also added to the SI), which generally falls close to expected 8-16 fold increase in wall clock time, while peak memory usage scales less rapidly at studied resolutions (4 fold increase per doubling of resolutions).

Kilometer-scale modeling in NeuralGCM. We concur with the reviewer that the current version of NeuralGCM would not scale to a kilometer-scale model. Scaling NeuralGCM to a km-scale model would require substantial resources to develop the model further, and it is likely not something that we could do in the near future. However, we show that NeuralGCM already competes with conventional forecast models despite its lower resolution, thanks to the learned physics. At the moment it does not

	NeuralGCM-2.8°	NeuralGCM-1.4°	NeuralGCM-0.7°	NeuralGCM-ENS (1.4°)
Training time	1 day	1 week	3 weeks	10 days
Device	16 TPU v4	16 TPU v4	256 TPU v4	128 TPU v5e
Parallelism	batch=16	batch=16	batch=16 x=2 y=2 z=4	batch=16 x=2 z=2 ensemble=2
Parameter count	14.5 M	18.3 M	31.1 M	11.5 M
Inference time on one TPU v4 core (10 day forecast)	2.5s	12.6s	119s	12.4s
Inference speed on one TPU v4 core (sim days/day)	350 000	69 000	7300	70 000
Peak memory usage during inference	255MB	1100MB	4177MB	1011MB

Table 1 Resource requirements for different NeuralGCM models trained on Google Cloud TPUs

require kilometer-scale (or even in the non-hydrostatic regime) to provide competitive weather forecasts/climate statistics. We acknowledge that kilometer-scale outputs are highly valuable, and even if NeuralGCM can produce accurate statistics at a coarse resolution, it may not meet the needs of many end users. Nonetheless, we remain optimistic that if NeuralGCM can achieve accurate statistics and weather forecasts at a coarse resolution (the primary goal of hybrid models), post-processing tools for statistical downscaling could be utilized to generate kilometer-scale predictions, as evidenced in recent studies (e.g., Mardani et al. [1]).

2.Ablations. The article could use more ablations to convince the reader about reasons for specific and somewhat curious design choices. The most significant is the loss function. The combination of terms, although some explanations are given, appears to be a mish-mash of things that just worked. Although this is not uncommon in ML, where the end sometimes justifies the means, it would be helpful to provide ablations for high-wavenumber-filtering, bias-correction, spectral-matching, multi-step-training, spectral cutoffs, etc. Additionally, it might be helpful to describe the computational cost of these complex loss functions. Other areas that could use ablations are the roles of the encoder and decoder. It is unclear, for example, how the encoder interacts with multi-step training. Similarly, why were certain normalization and scaling choices made versus other possible strategies?

Following the reviewer’s suggestion regarding the inclusion of more ablation tests we have now added a section in the supplementary information about ablation tests, where we test the effect of using different loss functions on the skill and spectrum of models. We note that considering the numerous modeling decisions we have made, we could not explore all possible parameters, but we think we have included the most important tweaks, which as the reviewer suggested are related to the loss function. Specifically, we have incorporated a comparison of skill (measured via RMSE) and the (zonal wavenumber) spectrum for the following simulations: (1) default NeuralGCM model, (2) model that was trained without the bias loss, (3) model that was trained without applying filtering to the MSE loss and (4) model that was trained without

spectral loss, bias loss, and filtering on the MSE, resulting in a standard MSE loss. As figure 1 shows (this figure is now also in the SI), all these variants of the default NeuralGCM model have a lower (better) RMSE, but the problem with these models is that their spectrum changes with time, which leads to blurring (Figs 1d and 2, figures included in the SI). We now discuss these results in section H6.

Regarding the question of computational overhead due to the additional loss terms, we have found that the overhead from including these losses is very limited. This is because the cost of calculating the gradient is dominated by the differentiation through the dynamical core, not by the actual loss function in the end.

3. Interpretability. An attractive property of hybrid models is their interpretability compared to full model emulators like GraphCast. However, it appears that a lot of the interpretability in NeuralGCM is lost by design, because of the decoder. Some commentary on how this could be avoided or done differently would be satisfying. I imagine this loss in interpretability would likely hurt the potential operational use of the model (for climate).

We agree with the reviewer that examining the outputs on pressure coordinates (after the decoder step) could, in principle, diminish some of the interpretability of NeuralGCM. Our intuition is that incorporating strong physics priors in the form of a dynamical core largely fixes the model representation of the atmosphere, whereas encoder and decoder simply address nuances such as imbalance of the initial conditions with respect to the dynamical equations and extrapolation strategies used by ECMWF when producing ERA5 data.

To support this, we have added Figs. 3,4 (these figures are now also in the SI) showing the differences between encoder/decoder components and direct interpolation between pressure and sigma levels is very small and most notable in areas like extrapolation to pressure levels below ground. We note that we diagnose precipitation minus evaporation fully in the model coordinates without using a decoder and still get meaningful results.

In future work we believe that encoder and decoder components could be completely replaced by more predictable procedures. One approach would be to train NeuralGCM-like models against model reanalysis (rather than fixed ERA5 representation), which would address both initialization shock and regridding challenges.

Furthermore, to highlight the ability to interpret the different tendencies in NeuralGCM we also included a new SI section on “Interpretability of learned physics tendencies” where we show in Fig. 5 tendencies from our learned physics module and the dycore.

4. Generalization. Although the results shown are very impressive, more work remains on proving NeuralGCM’s extrapolation and generalization properties. Could the model be tested on plausible future SST scenarios (within the AMIP framework). Could greenhouse gas concentrations be incorporated into the model? What about other climate change markers? Could there be a climate-invariant reformulation of NeuralGCM and its inputs? Would NeuralGCM benefit from training on future climate simulations?

Fig. 1 Accuracy and spectrum analysis for models using different loss terms and models trained on different rollout lengths. (a-c) RMSE normalized by the RMSE of the default NeuralGCM-2.8° model, as a function of prediction lead time for simulations utilizing various loss functions: the default NeuralGCM-2.8° loss function (sum of filtered-MSE, spectral and bias losses; orange), a model trained without the bias loss term (red), a model trained without the filter on the MSE loss (pink), and a model trained with simple MSE loss function (without the spectral loss, without bias loss, and without the filter on the MSE; blue) and a model trained only up to 12 hour rollouts (grey), for (a) geopotential at 500 hPa, (b) temperature at 850 hPa, and (c) specific humidity at 700 hPa. Panel (d) depicts the zonal spectrum of specific humidity at 850 hPa in the tropics (defined from 15S-15N), normalized by ERA5 spectrum for the aforementioned models (except for the model model that is trained up to 12 hours as the spectrum diverges at day 15) and also includes the ERA5 spectrum (black). Solid lines show 1-day predictions and dashed lines show 15-day predictions. The zonal spectrum is truncated at wavenumber 40 to facilitate a clearer comparison between the models.

Fig. 2 Case study on the impact of different loss functions on forecast snapshots. The plot illustrates the specific humidity at 700 hPa for 1-day, 5-day, and 10-day forecasts from each model across latitudes 30S-70N and longitudes 162-300 for ERA5, NeuralGCM-2.8° as well as NeuralGCM-2.8° variants trained without bias loss, without an MSE filter, and with a simple MSE loss (excluding spectral loss, bias loss, and MSE filter). All models are at 2.8° resolution (with ERA5 data regridded to match this resolution). All forecasts are initialized at 2021-08-25T00z.

Firstly, we want to emphasize that, similar to any other ML algorithm, we do not anticipate NeuralGCM’s parameterization to perform well in extrapolation. Consequently, we advise against relying on NeuralGCM for extrapolating to unseen conditions. Despite this, we recognize that extrapolation might be reasonable in case the distribution does not substantially change and we have expanded our analysis to include AMIP scenarios with increased SST (see Figs 6,7 (which are now added to the SI) and new SI section I4)

Testing future SST scenarios (within the AMIP framework). To assess NeuralGCM’s ability to simulate a warmer climate, we conducted experiments (refer to the new section in the supplementary material) using the AMIP protocol with SST increased by 1K, 2K, and 4K. Comparisons were made between NeuralGCM in warmed AMIP experiments and a physics-based model running with AMIP configuration +4K SST (CESM2, and for one plot MIROC6). The focus was on examining known responses to climate warming, such as increased upper tropospheric warming, polar amplification, and polar jet shift. While NeuralGCM responses exhibited similarities with physics-based models for lower warming values (+1K, +2K SST), a significant issue was identified with climate drift (less than 0.5K in 34 years), as discussed in Section I.4.2 and shown in Fig 6 (added to SI). For the larger warming (+4K SST) the response to warming is unrealistic, and there is a substantial climate of almost 2 degrees drift over 34 years. We also compared how the atmosphere adjusts to a warming (to test the transient response) and we have found that the global mean temperature in the atmosphere adjusts roughly on the same time scale as a physics based model (Fig. 7 (added to SI)).

Extending the generalization test for weather forecasting. We also extended the tests on how well the NeuralGCM model can predict weather in future

Fig. 3 Comparison of NeuralGCM-0.7° model states produced by the encoder and that of the pressure to sigma level interpolation scheme. (a) Visualization of temperature, u component of wind and specific humidity with full encoder and that with learned components disabled. (b) Vertical profiles for the same variables evaluated at (longitude \approx 238 and latitude \approx 37).

years with warmer climates. Previously, we have compared the skill of NeuralGCM to GraphCast for 5 unseen future years. We have trained a NeuralGCM model using only data from 1979-2000, and tested its performance on 21 future years (Fig. 8 (added to SI) and section I.4.1).

Could greenhouse gas concentrations be incorporated into the model?

Yes, there are a variety of ways in which greenhouse gas could be integrated into NeuralGCM. The easiest approach would be to prescribe greenhouse gas concentration and include it as an input to our neural network. Concentrations could be a global average, a reasonable simplification representing a well-mixed atmosphere with respect to that greenhouse gas. Another option is to treat greenhouse gas as a tracer, easily advected within the model. In principle, learning the tendencies of greenhouse gas is also possible, but it may require a more elaborate chemistry model to account for sinks and sources of greenhouse gas. However, incorporating greenhouse gas in the manners

Fig. 4 Comparison of NeuralGCM decoder and underlying sigma level to pressure level interpolation scheme. (a) Visualization of geopotential, temperature, u component of wind and specific humidity from ERA5, 2 day NeuralGCM-0.7° prediction processed with NeuralGCM decoder and 2 day NeuralGCM-0.7° prediction interpolated from sigma to pressure levels without learned components of the decoder. (b) Vertical profiles for the same variables evaluated around Tibetan plateau (longitude ≈ 86 and latitude ≈ 32).

described above will not be sufficient to conduct meaningful simulations with future concentrations of greenhouse gas, as it will require our model to extrapolate to unseen greenhouse gas concentrations. To enable simulations with increased greenhouse gas concentrations in a meaningful way we will need to incorporate a physics-based radiation scheme (or alternatively, learn from data that includes increased greenhouse gas concentrations). We believe this is a larger project more suitable for future work, which we have added into the discussion section: “For climate projection, NeuralGCM will need to be reformulated to enable coupling with other Earth system components (e.g., ocean, land), and integrating data on the atmospheric chemical composition (e.g., greenhouse gasses, aerosols). There are also research challenges common to current

Fig. 5 Comparison on tendencies produced by the dynamical core and the learned physics component of NeuralGCM-0.7°. All tendencies are computed on 2020-08- 24T00z. The plot shows σ level slices of temperature, zonal component of wind and specific humidity, as well as tendencies associated with the dynamical core and learned physics.

ML-based climate models [2], including the capability to simulate unprecedented climates (i.e., generalization), adhering to physical constraints, and resolving numerical instabilities and climate drift.”

What about other climate change markers? We are not sure that we correctly understood this comment. Our interpretation is that the reviewer is interested in whether NeuralGCM is capable of simulating known markers of climate change. As mentioned above, we examined certain climate change markers such as upper tropospheric warming, polar amplification, and polar jet shift (Figs. 6), and found that for +1K and +2K SST warming NeuralGCM does show these markers. We note that certain climate change markers such as sea-level rise cannot be currently simulated in NeuralGCM since we do not have an ocean model.

Could there be a climate-invariant reformulation of NeuralGCM and its inputs? Yes, in principle, a strategy similar to a previous preprint (Beucler et al.[3]) can be employed by using relative humidity instead of specific humidity and buoyancy (or other variables such as $T(z)-T(1000\text{hPa})$) instead of temperature. However, certain complications may arise, such as the need to transform inputs like SST

Fig. 6 Response of temperature and zonal wind to sea surface temperature (SST) warming. Left panels depict the zonal mean temperature response per degree of SST increase as a function of latitude and pressure, calculated as the difference between AMIP runs with increased SST to AMIP simulations averaged over 1981-2013: (a) CESM with a +4K SST warming, (b-d) NeuralGCM with +1K, +2K, and +4K SST warming, respectively. Right panels illustrate the corresponding zonal mean zonal wind response per SST degree increase for (e) CESM with a +4K SST enhancement and (f-h) NeuralGCM with +1K, +2K, and +4K SST warming, respectively. All NeuralGCM models used initial condition from April 10th 1980.

to achieve climate invariance (one option could be using SST minus the temperature at the lowest atmospheric level). It's important to note that the climate-invariant formulation of parameterizations has not been tested in online simulations to the best of our knowledge, and its potential impact on NeuralGCM's skill is unknown. Additionally, there is currently no known climate-invariant transformation for the outputs, posing a challenge if the model is intended to extrapolate to unseen climates. Following this comment we added to the following text to the SI: "In the future, NeuralGCM could improve its ability to generalize to different climates, even those it has

Fig. 7 Temporal dynamics of temperature changes in AMIP simulations with warmed sea surface temperature (SST). Panel (a) shows the 40-day 850hPa global temperature response to a +4K SST increase in MIROC6 (defined as the difference between AMIP +4K and the standard AMIP run; represented in orange) and NeuralGCM (in blue). Panel (b) presents the global mean temperature at 850hPa across various NeuralGCM AMIP scenarios, including standard runs and those with +1K, +2K, and +4K SST enhancements, adjusting the global mean by subtracting the corresponding SST warming. Panel (c) displays the global mean temperature at 850hPa for both the CESM AMIP standard run and the CESM AMIP scenario with a +4K SST increase, applying the same SST adjustment methodology. Panels b and c cover the period from 1981 to 2013, with 2013 being the final year for which CESM AMIP +4K SST data were available. All NeuralGCM simulations commenced from the initial condition on April 10th, 1980.

Fig. 8 Extrapolation of mid-range weather forecast skill over 21 years. Relative Root Mean Square Error (RMSE), normalized to the year 2001, for a NeuralGCM-2.8° model trained on 1979-2000 ERA5 data. Results are shown for (a) 500 hPa geopotential height, (b) 850 hPa temperature, and (c) 700 hPa specific humidity as a function of forecast lead time.

not trained on, by using a mix of strategies. While climate-invariant methods [3] can help, training NeuralGCM on data obtained from simulations (e.g., of warmer climates) and incorporating physically-based models (e.g., radiation schemes) will likely be necessary.”

Would NeuralGCM benefit from training on future climate simulations?

In principle, we could train NeuralGCM using simulations of future climate, potentially enhancing its generalizability. However, a potential challenge arises as training the parameterization on model data may result in a less skillful parameterization for future climates, considering it only learned from a model, compared to its ability to simulate the current climate. Furthermore, learning from conflicting datasets (i.e., reanalysis and simulations) could compromise the skill of the parameterization even

in the current climate. Following this comment we added to the following text: “In the future, NeuralGCM could improve its ability to generalize to different climates, even those it has not trained on, by using a mix of strategies. While climate-invariant methods [3] can help, training NeuralGCM on data obtained from simulations (e.g., of warmer climates) and incorporating physically-based models (e.g., radiation schemes) will likely be necessary.”

5. **Instability.** It is somewhat worrisome that some members of NeuralGCM blow up. One might expect that with a numerical dycore the model should be stable, especially in the hydrostatic limit. But this is not the case and it is unclear why.

We acknowledge and agree with the reviewer’s concern regarding the instabilities observed in NeuralGCM, recognizing that this issue poses a significant problem. Presently, we do not have a solution for this problem, and we transparently report the number of simulations that exhibited instability. The reviewer suggests that the utilization of a numerical dycore should enhance stability, but at this point, we lack sufficient evidence to support this claim, especially since we have not conducted extensive testing to compare with, for example, a purely data-driven approach. A prior study (Brenowitz et al. [4]) demonstrated that combining NN with an atmospheric model could lead to unstable modes, which might also be a problem in NeuralGCM (although online training might reduce the severity of this problem).

It is worth noting that the tests performed on NeuralGCM, such as 40-year runs using 37 different initial conditions, are more comprehensive compared to existing attempts, which often report success based on a single run. While we agree that addressing the instability issue is crucial, it is beyond the scope of what we were able to accomplish in the current paper.

Following this comment, and a request from a different reviewer, we have now added an SI section and a Figs. 9,10 (included in the SI) that discuss and show two case studies on instabilities for the NeuralGCM-0.7° and NeuralGCM-1.4° models.

Minor:

1. **Can some parallels be drawn to the rapidly exploding field of diffusion modeling?**

We are not sure that we can find clear parallels between ML diffusion-based models and NeuralGCM. One thing that comes to mind is that diffusion models aim to generate the probability distribution of the dataset, which is close to what we do in the stochastic model.

2. **Is multi-step training needed? Can the model be trained on fewer steps, shorter timescales?**

In response to this comment, we have now included a section in the SI on training using shorter rollout (section H.6.2) where we show that a model trained on 12 hour rollouts performs worse, but more importantly it is substantially less stable (Figure 1a-c). We note that we have also tried to train a model on a 1-hour rollouts (which is a single physics timestep in the 2.8 resolution models) and we found that this model is not even stable for a 15 day weather forecasting task.

3. **Can the loss function be posed more elegantly and simply?**

We started with simpler loss functions, which we progressively made more complex as motivated by improved results. To illustrate the benefits of our more complex loss setup, we have added ablation tests showing results with simpler losses in

Fig. 9 Case study on instability in the NeuralGCM-1.4° model. (a-d) The temperature, (e-h) 12 hour temperature change at 1000hPa (i-l) zoom into longitudes 20 to 80 and latitudes 75S to 45S and (m-p) the vertical structure of 12 hour temperature change at latitude 60S, for different simulation lead times: day 388 (left column), day 389 (second column), day 390 (third column) and day 392 (right column). The simulation was initialized at 09-18-2019.

Fig. 10 Case study on instability in the NeuralGCM-0.7° model. (a-d) The temperature, (e-h) 12 hour temperature change at 1000hPa (i-l) zoom into longitudes 280 to 320 and latitudes 20S to the equator and (m-p) the vertical structure of 12 hour temperature change at latitude 10.1S, for different simulation lead times: day 138 (left column), day 139 (second column), day 140 (third column) and day 155 (right column). The simulation was initialized at 08-29-2019.

(section H.6.1). A more elegant loss formulation was also part of the motivation for our stochastic training setup, which uses a simpler two component loss function based on CRPS.

4. Why are two ensemble members sufficient for the stochastic training? Is there a mathematical explanation?

We have now added additional explanations in the SI where we write: “During training, for each initial time t , and forecast time $t+\tau$, we use one observation $Y(t+\tau)$ and exactly two forecasts $X(t+\tau)$ and $X'(t+\tau)$. This is the minimum needed for an unbiased CRPS estimate (??). It is possible to construct a CRPS loss with more than two forecast samples for every observation. As it turns out, the compute budget (for each minibatch) is better spent on greater variety of initial times. That is, rather than increasing ensemble size, we select a new observation at a new initial time, $Y(s+\tau)$ for $s \neq t$, and once again create the minimum number of forecasts $X(s+\tau)$, $X'(s+\tau)$. This is due to the fact that while adding more ensemble members reduces the variance contribution of forecasts, the variance due to having one single observation is fixed. The result of adding more forecasts would be a sub-linear variance reduction. On the other hand, increasing the number of distinct observations in a minibatch gives a linear reduction in variance.”

5. Can the numerical dycore be done away with? Perhaps using a lightweight ML surrogate?

In principle, yes. This would be a potentially fruitful area to explore in future work. To our understanding the reviewer asked about having a different ML component to emulate only the dycore, which we think is likely doable based on the success of ML-only weather forecasting models for the full atmosphere. We note that such a strategy (using ML instead of the dycore) might have some disadvantages/complications. For instance, if attempting to teach a new ML component the dynamical core tendencies through online learning with ERA5 data, maintaining a clear separation between physics and dycore becomes challenging. The ‘physics’ neural network could potentially predict some of the dycore tendencies. One feasible approach to address this is by initially training an ML model to mimic a dycore, a process that would involve generating data that includes a model with only a dycore. Another concern is that running an interpretable ML dycore would require a time-step size that is comparable to our physics time-step (~ 30 minutes). This would significantly reduce or eliminate the speed advantages of ML-only models such as GraphCast, which are often facilitated by their larger time-step size (~ 6 hours). We note that we verified that we could train an ML-only model within the NeuralGCM modeling framework (albeit with the single column parameterization structure), and it was stable for weather prediction tasks although less skilled (Fig. 11 which is now added to the SI; although we acknowledge that this pure ML-approach is not exactly what the reviewer asked about).

6. Isn't the spectral loss the same as the L2 loss in physical space (from generalized Parseval's theorem)?

The spectrum MSE loss discussed in Appendix G.4.2 is different from the L2 loss in physical space because it aggregates the norm of spherical harmonic coefficients for each total wavenumber prior to computing the loss. In other words, spectrum MSE loss is not sensitive to mixing of spherical harmonics with different zonal wavenumbers

Fig. 11 Learning curves comparing the hybrid NeuralGCM model with the ML-only variant of NeuralGCM. RMSE as a function of the number of training years (with all models trained on the years leading up to 2017 and evaluated on data from 2018, utilizing varying lengths of training data) for both NeuralGCM-2.8° (blue) and its ML-only counterpart, NeuralGCM-ML-only-2.8° (orange), which is similar to NeuralGCM-2.8° but without the dynamical core. The RMSE values are presented for forecasts at day 3 for (a) geopotential at 500hPa, (b) temperature at 850hPa, and (c) specific humidity at 700 hPa.

corresponding to the same total wavenumber. We agree that if such aggregation was not performed then imposing L2 loss on all of the spherical harmonics coefficients individually would be equivalent (modulo the truncation error) to the L2 loss in physical space.

7. How does time-step filtering and the other loss functions affect the effective resolution of the model?

We are not 100% sure what the reviewer refers to as the ‘time-step filtering’? If the reviewer is referring to the filtered MSE, the utilization of filtered MSE increases the effective resolution compared to using MSE loss as demonstrated in Figs.1,2 (added to SI and discussed in new section about ablation tests). When assessing effective resolution we examine the smallest scale features that the model maintains beyond the time horizon that was used for training. If the reviewer is alluding to the exponential filters used in our model, it is accurate that the usage of such filters can potentially impact the effective resolution of models. However, it is important to note that employing filters is a common approach in dycores to enhance stability. For instance, hyperdiffusion, which is commonly used in dycores, may potentially decrease the effective resolution of the model. We would like to emphasize that one of the reasons for extensively presenting the spectrum of our model is to facilitate readers in examining the spectrum.

8. Can NeuralGCM be reformulate to predict both precipitation and evaporation?

In principle, yes. That could be done in many different ways. For example, we could use a conventional model for surface fluxes to calculate evaporation and then calculate the precipitation by adding to P-E the evaporation obtained from the surface fluxes. This would effectively extend the dycore to include surface evaporative fluxes. Another approach would be to train an NN that would predict evaporation (potentially optimized to predict the same fluxes as in ERA5 data). We now discuss this in SI section H.5) where we write: “In this work, we only diagnosed precipitation minus evaporation from NeuralGCM. However, in future work, we plan to develop a scheme to reformulate NeuralGCM to predict precipitation and evaporation separately. This could be achieved, for example, by using conventional parameterization to estimate evaporation (and then calculating precipitation by adding it to P-E). Another approach could involve training a neural network (NN) specifically to predict evaporation, potentially optimized to predict the same fluxes as those in ERA5 data.”

9. Why does the model under-predict extremes in the tropics?

We do not yet any have a satisfactory explanation for this, which is why we do not attempt an explanation in text. One possibility is that the model was not sufficiently encouraged to achieve the correct distribution, as we did not add any direct loss term on P-E. Another possibility is that the deterministic loss does not sufficiently promote the prediction of extreme events. As a side note, we wish to highlight that one reason we did not attempt to accurately emulate the ERA5 P-E distribution is its known inaccuracies. In future work, we hope to emulate observational-based precipitation data rather than relying on ERA5.

10. Why does the model have grid-scale artifacts (Fig. I-26)?

We are uncertain about the origin of the artifacts observed in the lower-resolution models, specifically in the variable P-E (Precipitation minus Evaporation) as other

variables do not exhibit similar noise. It is worth noticing that P-E is calculated in sigma coordinates, and therefore it is not decoded. We suspect that the noise in P-E arises because, during the calculation of neural network (NN) physics tendencies, we transform the predicted tendencies from a Gaussian grid representation to a spectral representation. Subsequently, we perform the inverse transformation, from spectral representation to Gaussian grid representation, before calculating P-E. Another possibility (which was also raised by another reviewer though we are not convinced yet that this is the reason) is that the NN parameterization we use compensates for errors in the advective tendencies of the dycore (so the tendencies obtained from the NN parameterization might include tendencies that are associated with horizontal advection). Currently, we could not solve this grid-scale noise.

Following this comment we have added the following text to the SI: “We note that the calculation of $P - E$ assumes that the dynamical core is responsible for all horizontal motions; thus, $P - E$ could be diagnosed using Eq. ???. However, the physics module might also learn to correct errors originating from inaccuracies in the dynamical core (e.g., as a result of calculating advective tendencies on a coarse grid), which may introduce errors into the calculation of $P - E$. Given that $P - E$ calculated from NeuralGCM-0.7° appears generally consistent with ERA5 in the weather forecasting scenario (Fig.??), this suggests that any error is likely small. However, at lower resolutions, the error could be larger. For example, Fig ?? displays the climatology of $P - E$ calculated with NeuralGCM-1.4°, which shows clear artifacts, although it remains unclear whether these artifacts stem from the aforementioned error in calculation or another issue.”

11. Why does the model have a larger spread than AMIP runs, even close to the surface where temperatures are expected to be more constrained by SST?

We do not have an answer to this question. It is possible that this reflects some problematic features of NeuralGCM. It is important to note that we acknowledge this potentially problematic feature of our model in the text: “NeuralGCM does show a wider spread in its predictions than the AMIP runs, even at levels near the surface where temperatures are typically more constrained by prescribed SST.”

4 Reviewer 2

The paper presents a novel, fully differentiable hybrid ML-physics general circulation model. This is indeed impressive work. It addresses the instability of long rollouts in pure-AI weather prediction models by incorporating explicit large-scale physics. Additionally, it overcomes the limitations of traditional hybrid ML-physics climate modeling, critically hindered by non-differentiable Fortran codebases. The formulation of the domain knowledge-infused loss function is particularly creative and effective. The manuscript is generally clear, thoroughly incorporating supplementary materials. However, there are several areas where clarity could be enhanced:

We thank the reviewer for the careful evaluation of our work and useful comments.

1. The novelty and contribution of this work seem understated. The authors should more clearly articulate the work’s impact and contribution in the introduction.

We have now made sure that the last paragraph in the introduction summarizes the contribution of this work.

2. The paper does not adequately address the limitations of this work. It gives the impression that this model solves all climate modeling problems by breaking the trade-offs between computational cost and accuracy. A more detailed discussion of limitations, perhaps referring to the 'SILIENT' principle in Bretherton (2023), would be beneficial.

Following the suggestion of the reviewer, in order to highlight the limitations of NeuralGCM, we now discuss the limitations of NeuralGCM in the discussion where we write: “For climate projection, NeuralGCM will need to be reformulated to enable coupling with other Earth system components (e.g., ocean, land), and integrating data on the atmospheric chemical composition (e.g., greenhouse gasses, aerosols). There are also research challenges common to current ML-based climate models [2], including the capability to simulate unprecedented climates (i.e., generalization), adhering to physical constraints, and resolving numerical instabilities and climate drift.”

In the SI we also added the following text: “In the future, NeuralGCM could improve its ability to generalize to different climates, even those it has not trained on, by using a mix of strategies. While climate-invariant methods [3] can help, training NeuralGCM on data obtained from simulations (e.g., of warmer climates) and incorporating physically-based models (e.g., radiation schemes) will likely be necessary.”

3. I suggest the authors provide recommendations for future research, outlining both immediate and long-term steps for those inspired by this work.

As mentioned above, we now discuss what would be needed for further improving NeuralGCM. Furthermore, as mentioned in the previous comment we highlight the challenges that need to be addressed before NeuralGCM can be more useful for climate modeling.

4. A more detailed visual presentation of the NGCM is needed (maybe in the SI?). While Figure 1 shows the overall structure, it lacks details on components like the encoder, decoder, and learned physics (spatial embeddings and block architecture).

Following the suggestion of the reviewer we have now added to the SI detailed figures showing the structure of the encoder, decoder and the learned physics (Figs 12,13).

Fig. 12 Visualization of the data flow in encoder and decoder modules of NeuralGCM.

5. A complete description of input and output variables is missing. A table summarizing all input, output, and forcing variables would be helpful. This is especially helpful since the forcing variables are usually treated as input variables in similar works.

Following the suggestion of the reviewer we now describe in Fig.13 the inputs and outputs of the network we use.

6. The comparison of NGCM with XSHIELDS and AMIP models (L443-454 and L502-514) is a bit one-sided. These models are built on general atmospheric physics principles, unlike NGCM, which is explicitly trained to emulate ERA5. The nuanced differences, especially the lesser tuning of XSHIELDS due to computational costs, should be highlighted.

Fig. 13 Visualization of the data flow in the learned physics module of NeuralGCM.

We appreciate the reviewer’s comment and we have now added the following text: “This comparison slightly favors NeuralGCM because NeuralGCM was tuned to match ERA5, but the discrepancy between ERA5 and the actual atmosphere is small relative to model error.”.

7. Regarding the stability of hybrid climate modeling (L498-500), including common failure modes in the SI would be beneficial for hybrid model developers.

Following this comment, we have now added an SI section H.7 and a Figs. 9, 10 that discuss and show two case studies on instabilities for the NeuralGCM-0.7°, NeuralGCM-0.7°.

8. Information about the model code repository and its accessibility is missing. Is it going to be open-sourced?

Yes, we have made our dynamical core and trained NeuralGCM models available on github in github.com/google-research/dinosaur and github.com/google-research/neuralgcm repositories correspondingly. We have also added data and code availability section to the main paper.

9. Several important references are missing and should be incorporated: a. Pathak, Jaideep, et al. “Fourcastnet: A global data-driven high-resolution weather model using adaptive fourier neural operators.” arXiv preprint arXiv:2202.11214 (2022). b. Nguyen, Tung, et al. “ClimaX: A foundation model for weather and climate.” arXiv preprint arXiv:2301.10343 (2023). c. Watt-Meyer, Oliver, et al. “ACE: A fast, skillful learned global atmospheric model for climate prediction.” arXiv preprint arXiv:2310.02074

(2023). d. Han, Yilun, Guang J. Zhang, and Yong Wang. "An ensemble of neural networks for moist physics processes, its generalizability and stable integration." *Journal of Advances in Modeling Earth Systems* 15.10 (2023): e2022MS003508. e. Hakim, Gregory J., and Sanjit Masanam. "Dynamical Tests of a Deep-Learning Weather Prediction Model." arXiv preprint arXiv:2309.10867 (2023).

We thank the reviewer for pointing out these references. Following this comment we now cite Watt-Meyer, Oliver, et al. "ACE: A fast, skillful learned global atmospheric model for climate prediction." arXiv preprint arXiv:2310.02074 (2023) [5]. In the introduction we now write: Furthermore, although there have been some successes in using ML approaches on longer time scales [5, 6], ML models have not demonstrated the ability to outperform existing GCMs."

We also now cite: Han, Yilun, Guang J. Zhang, and Yong Wang. "An ensemble of neural networks for moist physics processes, its generalizability and stable integration." *Journal of Advances in Modeling Earth Systems* 15.10 (2023): e2022MS003508 [7].

We decided not to include the first two papers the reviewer suggested since we do not find them as important/innovative as other papers we cite (e.g., the skill of Fourcastnet is substantially lower than Pangu/GC). We also do not include the last paper suggested by the reviewer because it feels incomplete without comparisons between Pangu and a physics-based model, and we are concerned about the validity of its test-cases (modifying geopotential independently of temperature violates hydrostatic balance). We are also very close to the maximum limit of 50 citations.

10. The order of figure references is not consistent (e.g., L437-442). This should be corrected throughout the manuscript.

To maintain the organization and readability of the appendix, we decided not to reorganize the order of the figures.

11. The title is too vague, not efficiently conveying the topic of the manuscript.

Following this comment we have now updated the title of the manuscript. The current title is: "Neural General Circulation Models for Weather and Climate"

12. Minor comments: a. More description is needed for regridding methods beyond "conservative."

We have clarified in the training section that we are using conservatively regridded ERA5 data where we refer to appendix section G.2 that elaborates on conservative regridding.

b. L189 vs. L208: There's inconsistency in the rollout schedule

We have added a clarification to the discussion of the stochastic model training that the loss was computed on rollout lengths up to 5 days. The 3-day rollout on L208 refers to deterministic models which were trained only on up-to 3 day rollouts.

c. L212-214: Clarification on how randomness is incorporated

We have now modified this paragraph to explain how the randomness is incorporated into NeuralGCM where we now write: "Stochastic NeuralGCM models incorporate inherent randomness in the form of additional random fields passed as inputs to neural network components."

We also reference there an SI section about the additional input features that we use for stochastic models.

d. L263-264: Generalizability compared to other models and variables/levels

We were not sure what the reviewer was referring to.

e. L271: What about 0.7deg version?

We have not yet trained a stochastic 0.7deg model due to the large computational expense.

f. Fig 2: Clarification of grey areas is missing

Done (they are regions below the surface)

g. L1017-2020: How were the land surfaces treated?

We now clarified this sentence and we write: “When running seasonal and climate evaluation forecasts, we do prescribe the SST and sea ice concentration from ERA5 data and update them at 6-hour intervals for the 2.8° resolution model and 12-hour intervals for the 1.4° resolution model (details on how land surfaces were used can be found in ??). This simulates a one way coupling to an ocean model that follows historical evolution.”

h. Fig H3: What makes GraphCast more accurate for a short lead time? Or, What makes NGCM less accurate for a short lead time?

We think that the most probable reason is the higher resolution of GC compared to NeuralGCM. In addition to being almost 3 times coarser in each latitude/longitude direction, NeuralGCM also projects the atmospheric state onto the basis of spherical harmonics. While GraphCast potentially takes advantage of high resolution data to make more accurate predictions at short lead times, at longer lead times, when GraphCast makes blurry predictions, the effective model resolution is quite similar (see power spectra shown in Fig. H16). Consistently we find that a significant improvement in short lead time performance as the resolution of NeuralGCM increases, as shown in Fig. H5. It’s also worth noting that the stochastic NeuralGCM model introduces perturbations to the initial state which affects the performance at short lead times more substantially.

When going to higher resolution there are clear benefits at shorter lead times. This is also seen in Neural GCM. Fig. H5 shows as we increase the resolution of our model there is a substantial decrease in the loss in the first few days. Furthermore, for the deterministic model it is not clear that GC is more accurate on short lead times beyond 12 hours. The fact that GC is more skilled in 12 hours is possibly because GC at short lead times has a high effective resolution which can capture the high wavenumbers (which deteriorates rapidly as the model becomes blurry). However, when designing the loss function for NeuralGCM, we aimed for the NeuralGCM spectrum to be consistent across time. Therefore, when initializing the model, we did not demand the model to get correctly wavenumbers that are beyond (what we thought of) its effective resolution.

Following this comment, we added the following text to the SI: “We note that NeuralGCM-0.7° is almost 3 times coarser than GraphCast, and it is likely that higher-resolution NeuralGCM models would improve their short lead time predictions, a trend already observed as NeuralGCM models ??.”

i. Figs H7 and H8: Is there any explanation for common behaviors (e.g., dipping and recovering CRPS around 3-5 day’s mark; the initial increase of the spread-skill ratio)?

The most probable reason that stochastic NeuralGCM model has these behaviors are related to the noise we introduce to the encoder in order to perturb initial conditions. We note that all these plots are showing relative RMSE (compared to IFS-ENS), and these results imply that at early times the RMSE (see Figure 2 in the main text) and CRPS of Neural GCM is larger, however we note that RMSE (and also CRPS) at early times is very small so the absolute difference is very small between these two predictions. We note that NeuralGCM-ENS was optimized on 5 day rollout, and the perturbation that are added to the initial conditions likely improve the prediction at later times, while adding some small error (in absolute terms) at earlier times (but overall these perturbation likely decrease the loss). Another possible contributing factor to these results is that the NeuralGCM is substantially less than IFS resolution. So in principle, IFS runs can add initial perturbations on a smaller scale (but still for example, match IFS-analysis at a coarser scale), while any initial perturbation of NeuralGCM model will cause an increase in RMSE. We have now included the following text in the SI: “ECMWF-ENS has better scores than NeuralGCM-ENS at very early lead times. However, both RMSE and CRPS values at these early times are small, indicating a negligible absolute difference between the models. ECMWF-ENS’s advantage possibly stems from two factors: a) NeuralGCM-ENS was optimized for 5-day rollouts, and initial condition perturbations likely improve later-stage predictions while introducing minor errors (in absolute terms) at earlier stages. Overall, these perturbations likely reduce loss. b) NeuralGCM’s lower resolution (1.4°) compared to ECMWF-ENS (0.2°) limits its ability to incorporate fine-scale initial perturbations. Namely, ECMWF-ENS can introduce these perturbations while still matching IFS-analysis at a coarser scale, whereas perturbations in NeuralGCM would always increase RMSE.”

5 Reviewer 3

A. Summary of the key results This study presents the first general circulation model which successfully uses a neural network for all subgrid-scale “physics” to enable long accurate simulations. The key methodological advance over previous works is the use of a differentiable dynamical core which allows optimization accounting for the combined effect of dynamics and the learned physics over multiple time steps. The resulting system is able to make very accurate weather forecasts when initialized from ERA5 reanalysis and also has attractive ensemble prediction capabilities. Impressively, it is also capable of 30-year simulations with low climate biases, although about one third of simulations are unstable when run for this length.

We thank the reviewer for the careful reading and very useful comments.

B. Originality and significance The successful use of a differentiable dynamical core, and the combined results of excellent weather forecast skill and mostly small climate biases are certainly novel and significant. However some prior works should be acknowledged. For example, when discussing previous hybrid dynamics + ML physics approaches, the authors do not cite a line of work that has successfully emulated cloud resolving models embedded in a GCM with realistic topography. E.g. see <https://doi.org/10.1029/2020ms002076> and <https://doi.org/10.1029/2022ms003508>.

We thank the reviewer for pointing out these papers. Following the suggestion of the reviewer we now cite now: Han, Yilun, Guang J. Zhang, and Yong Wang. “An ensemble of neural networks for moist physics processes, its generalizability and stable integration.” *Journal of Advances in Modeling Earth Systems* 15.10 (2023): e2022MS003508 [7].

The first paper the reviewer outputted (<https://doi.org/10.1029/2020ms002076>) does not show results for online (prognostic) simulations beyond a single column model simulation, and therefore we do not think it is relevant enough to cite here. Furthermore, given that we now cite 3 different papers that have aimed at using hybrid models with realistic topography, we do not think that adding this citation is necessary.

Furthermore, the authors claim that pure ML models trained to predict weather are not capable of long-term stable simulations (lines 35-37 and 83-84). While it is true that the existing ML models which beat ECMWF’s IFS model do not show forecasts beyond 14 days, promising results for year long forecasts were shown by <https://doi.org/10.1029/2020MS002109> and <https://arxiv.org/abs/2306.03838>. Additionally, stable and accurate 10-100 year forecasts from a pure ML model were shown by <https://doi.org/10.48550/arXiv.2310.02074> (albeit trained on climate model output, instead of ERA5). These prior results should be acknowledged.

Following the reviewer’s comment we now modified the related text. In the introduction and we now cite two of the papers the reviewer’s referred to (since we have a limit of 50 citations, we would not be able to accommodate more citations) and we now write: “Furthermore, although there have been some successes in using ML approaches on longer time scales [5, 6], ML models have not demonstrated the ability to outperform existing GCMs.”

In the abstract we have not changed the text since we refer there to ML approaches that are competitive then existing approaches for weather forecasting (and these ML

models have not shown stability properties that they could be used for simulations of climate)

C. Data methodology In general the approach is appropriate given the goals of the study. However, it should be made clearer what the advantage of the hybrid dynamics + ML approach is over a pure ML strategy. For example, is the improved performance over prior approaches (say, in terms of 5-15 weather forecast skill or sharpness of longer forecasts) due to the hybrid modeling approach or is it due to the various modifications of the loss function, which could easily be applied to a pure ML model?

We agree with the reviewer that ML approaches could use the same loss function and potentially improve the ability to simulate a more realistic spectrum. Furthermore, we have not demonstrated (and therefore did not make any such claim) that “pure” ML approaches could not achieve similar results.

Following this comment of the reviewer we added the following text to SI: “We note that the loss functions used in NeuralGCM, which promoted a more realistic spectrum and the ability to produce ensemble forecasts, could also be applied in ”pure” ML approaches, which we expect would likely improve such models as well.”

In my opinion, the advantages of the hybrid approach are interpretability, and possibly improved generalizability and a reduction in the amount of training data required. However these strengths are only tangentially discussed which is unfortunate.

We agree with the reviewer that the strengths of hybrid models include interpretability, and possibly improved generalizability, and a reduction in the amount of training data required. Additionally, we believe that an essential advantage is that hybrid models are more likely to be extensible (e.g., easier to add non-ML components and couple to other models). For instance, NeuralGCM models could incorporate physically-based radiation schemes, a task more challenging with pure ML models due to reasons such as their large time step. Following the reviewer’s comment we have now added text and figures to highlight these in the text:

Reduction in the amount of training data required. We have added an SI section (section H.6) that includes ablation tests. One of the ablation tests aims at investigating whether the dycore reduces the amount of training data required. Specifically, we have tested this idea within the NeuralGCM modeling framework by training ”full” NeuralGCM models (dycore + NN) as described in the manuscript and NeuralGCM-pure-ML models (trained without the dycore and all other elements are the same). In a new Fig. 11 in the SI we show the evaluation of 3-day errors in these models (when they are trained with different data volumes). While we find that the dycore enhances performance (which is not surprising given the single-column parameterizations we use), we do not find that the full model needs less data for improving its skill. We speculate that the single-column approach by itself may reduce the amount of data necessary to train models. Unfortunately, making decisive and general claims is challenging since comparing NeuralGCM to the variety of existing ML approaches is difficult. Even if we could demonstrate that NeuralGCM can learn using less data than, say, GraphCast, we are uncertain whether the same conclusion would hold for a different ML architecture.

Generalizability. We have now added an SI section discussing the ability of NeuralGCM to generalize to warmer climates (Section I.4). Specifically, to assess NeuralGCM’s ability to simulate a warmer climate, we conducted experiments (Section I.4.2) using the AMIP protocol with SST increased by 1K, 2K, and 4K (Fig. 6). Comparisons were made between NeuralGCM in warmed AMIP experiments and a physics-based model running with AMIP configuration +4K SST (CESM2, and for one plot MIROC6). The focus was on examining known responses to climate warming, such as increased upper tropospheric warming, polar amplification, and polar jet shift. While NeuralGCM responses exhibited similarities with physics-based models (Fig. 6) for lower warming values (+1K, +2K SST), a significant issue was identified with climate drift, as discussed in Section I.4.2 and shown in Fig. 7. For the larger warming (+4K SST) the response to warming is unrealistic, and there is a substantial climate of almost 2 degrees drift over 34 years. We also compared how the atmosphere adjust to a warming (to test the transient response) and we have found that the global mean temperature in the atmosphere adjust roughly on the same time scale as a physics based model (Fig. 7a).

We stress that we remain unsure whether NeuralGCM in its current structure could enhance generalizability compared to “pure” ML models as we have not directly compared it to pure ML models (and also SOTA ML models currently cannot run AMIP-like configurations at this moment to the best of our knowledge). However, we believe that incorporating a radiation scheme and other physically based parameterization within NeuralGCM could help with generalization, and incorporating such physics-based models with pure ML-approaches will be probably more challenging. We note that the main distribution shift in warmer/cooler climates occurs in the thermodynamic and moisture fields, primarily modeled through NN parameterization. Therefore, similar to any other ML algorithm, we do not expect NeuralGCM’s parameterization to excel in extrapolation. However, we found that extrapolation was reasonable when the distribution does not substantially change (e.g., +1K, +2K SST).

Interpretability. We believe that we have demonstrated a level of interpretability by diagnosing precipitation minus evaporation (and thus adhering to the water budget). Furthermore, we have added a section (SI section C.6) on “Interpretability of learned physics tendencies” (see below)

In terms of the interpretability question, it would be useful to show some of the internal workings of NeuralGCM. For example, do the “dynamics” and “physics” tendencies in a single time step (Fig. 1) correspond to what we would expect in a typical GCM? In particular, it seems highly likely that the learned part of the tendency (which is called “physics”) actually represents some combination of subgrid-scale processes and a correction for errors introduced by the explicit dynamics. This interpretation is supported by occurrence of grid-scale noise in the P-E plots shown in Figures H21d and I26d, and it should be discussed in the main text.

Following the suggestion of the reviewer we added a section (SI section C.6) on “Interpretability of learned physics tendencies” and a plot (Fig. 5) showing the physics tendencies and dynamical core tendencies from a single time step.

We are uncertain about the origin of the artifacts observed in the lower-resolution models, specifically in the variable P-E (Precipitation minus Evaporation) as other

variables do not exhibit similar noise. We appreciate the interpretation of the reviewer that they are a result of a correction of the advective tendencies by the NN, Following this comment we we have now added this text to the SI: “We note that the calculation of $P - E$ assumes that the dycore is responsible for all horizontal motions; thus, $P - E$ could be diagnosed using Eq. ???. However, the physics module might also learn to correct errors originating from inaccuracies in the dycore (e.g., as a result of calculating advective tendencies on a coarse grid), which may introduce errors into the calculation of $P - E$. Given that $P - E$ calculated from NeuralGCM-0.7° appears generally consistent with ERA5 in the weather forecasting scenario (Fig.??), this suggests that any error is likely small. However, at lower resolutions, the error could be larger. For example, Fig ?? displays the climatology of $P - E$ calculated with NeuralGCM-1.4°, which shows clear artifacts, although it remains unclear whether these artifacts stem from the aforementioned error in calculation or another issue. “

However, we are not sure whether this is the underlying reason for inaccuracies in the calculation of P-E. We suspect that the noise in P-E arises because, during the calculation of neural network (NN) physics tendencies, we transform the predicted tendencies from a Gaussian grid representation to a spectral representation. Subsequently, we perform the inverse transformation, from spectral representation to Gaussian grid representation, before calculating P-E. Furthermore, it is worth noticing that P-E is calculated in sigma coordinates, and therefore it is not decoded.

D. Appropriate use of statistics and treatment of uncertainties Yes. The analysis of ensemble forecast skill is nice to see.

E. Conclusions: robustness, validity, reliability Many aspects of methodology (e.g. loss function adjustments which improve 5-15 day skill and sharpness of forecasts) could easily be applied to a pure-ML forecasting system. This should be acknowledged, and the particular advantages of the hybrid physics-ML system should be clarified. Additionally, it is implied that this approach could be useful for climate change simulations. How can we trust it for future or past “out of training distribution” conditions when it is only trained on historical reanalysis data?

Following the comments of the reviewer we now acknowledge in the SI that: “We note that the loss functions used in NeuralGCM, which promoted a more realistic spectrum and the ability to produce ensemble forecasts, could also be applied in “pure” ML approaches.”

Regarding climate change simulation, we now write in the discussion: “For climate projection, NeuralGCM will need to be reformulated to enable coupling with other Earth system components (e.g., ocean, land), and integrating data on the atmospheric chemical composition (e.g., greenhouse gasses, aerosols). There are also research challenges common to current ML-based climate models [2], including the capability to simulate unprecedented climates (i.e., generalization), adhering to physical constraints, and resolving numerical instabilities and climate drift.”

We also added to the SI: “In the future, NeuralGCM could improve its ability to generalize to different climates, even those it has not trained on, by using a mix of strategies. While climate-invariant methods [3] can help, training NeuralGCM on data obtained from simulations (e.g., of warmer climates) and incorporating physically-based models (e.g., radiation schemes) will likely be necessary.”

F. Suggested improvements: experiments, data for possible revision More timing information for inference speed should be provided (currently only the 1.4° timing is mentioned). Showing more snapshots of the “physics” tendencies would help with understanding the characteristics of the model. A stronger generalization test would be useful for showing the strengths of the hybrid ML approach. E.g. try applying a +2K or +4K sea surface temperature perturbation.

Following the suggestions of the reviewer (and as described in previous comments), we have now included snapshots of the physics tendencies of the model, ran AMIP experiments with warmer SST and added a table (table 1 which was also added to the SI).

G. References: appropriate credit to previous work? Some additional references that should be cited were listed in point 2 above.

Following the suggestion of the reviewer we now include 2 of the citations the reviewer suggested.

H. Clarity and context The writing and presentation is very clear. The appendix material is somewhat overwhelming in quantity, but useful for understanding the various strengths/weaknesses of NeuralGCM.

Thanks!

Below I provide a series of comments noted while reading the manuscript.

Line 36-38 and 83-84: previously published ML-based models do show promising results for long-term prediction (e.g. Weyn et al. 2020, Bonev et al. 2023, Watt-Meyer et al. 2023). Although none of those models are also able to have state-of-the-art weather forecast skill, it is not clear that this is necessary for them to be useful for climate prediction. These prior works should be acknowledged.

We now acknowledge this issue in the introduction, and we also now cite Weyn et al. 2020 [6] and Watt-Meyer et al. 2023 [5].

In the abstract we have not changed the text since we refer there to ML approaches that are competitive then existing approaches for weather forecasting (and these ML models have not shown stability properties that they could be used for simulations of climate)

Line 120-122: the SPCAM emulator work has been fairly successful in realistic geography setting.

We now have now added a citation that uses a hybrid model based on SPCAM emulator in realistic geography setting (Han, Yilun, Guang J. Zhang, and Yong Wang. “An ensemble of neural networks for moist physics processes, its generalizability and stable integration.” *Journal of Advances in Modeling Earth Systems* 15.10 (2023): e2022MS003508) [7]

Lines 136-142: although it is called a “physics” module, if the goal is to replicate ERA5, the ML-learned component will also have to correct errors from the relatively coarse dynamical core. This needs to be stated up-front. (See also lines 986-987)

Following this comment, we have now edited the text in the SI to: “In NeuralGCM, effects of physical processes not accounted for by the dynamical core, as well as computational errors within the dynamical core, are approximated by neural networks.”

And in the main text we now write to: “The learned physics module predicts the effect of unresolved processes such as cloud formation, radiative transport, precipitation and subgrid-scale dynamics, on the simulated fields using a neural network.”

Lines 156-157: the fact that horizontal gradients help prediction is a hint that some part of the “physics module” target is a dynamics correction. The grid-scale noise in Figures H21d and I26d also indicates the same thing.

As discussed in previous comments we now acknowledge that possibility. We note that a previous paper has demonstrated that using non-local information in physics parameterization could improve physics parameterization (see Wang et. al. 2022[8]), so we think it is not clear that the fact that adding the horizontal gradients as inputs to the physics module improves the skill necessarily means that the NN uses them to correct the dynamical core or it mainly helps to estimate the effect of unresolved physical processes.

Lines 161-163: encoder-decoder is cool!

Thanks

Lines 179-181: usually this is just done for radiation, not all physics.

In the IFS model the substepping is only for radiation, but in climate models sometimes the physics and dynamics have different time steps. E.g., in CAM5 documentation it is written: “Since the scheme is explicit and restricted to small time-steps by its non-advective component, it sub-steps the dynamics multiple times during a longer parameterization time step.” (*cam5_desc.pdf*)

Lines 395-400: this is a somewhat weak generalization test (although admittedly one that GraphCast does not pass). What happens if you do something like apply a +2K SST perturbation to 2020?

We agree with the reviewer and therefore, as discussed above we have now included perturbed AMIP simulations (34 year runs of +1, +2, +4K SST warming). Furthermore, we also extended the tests on how well the NeuralGCM model can predict weather in future years with warmer climates. Previously, we have compared the skill of NeuralGCM to GraphCast for 5 unseen future years. We have trained a NeuralGCM model using only data from 1979-2000, and tested its performance on 21 future years ((Fig. 8) and section I.4.1).

Line 423-424 and lines 498-499: what is the nature of the instabilities that sometimes occur?

Following this comment we have now included a section in the SI (section H.7) that discusses and shows two case studies of instabilities (see Figs. 9,10).

Figure 4b: Is this also 850hPa temperature? It seems there is more ensemble spread in NeuralGCM early in the simulation—is this true, and if so do you have any idea why?

It is 850hPa. We do not have an answer to this question. It is possible that this reflects some problematic features of NeuralGCM. It is important to note that we acknowledge this potentially problematic feature of our model in the text: “NeuralGCM does show a wider spread in its predictions than the AMIP runs, even at levels near the surface where temperatures are typically more constrained by prescribed SST.”

Lines 531-532: it would be useful to report the timings for other resolutions also.

We have now added table 1 (also added to the SI) showing the timings for all resolutions.

Lines 553-555: do you have any ideas for ways to resolve these generalization challenges? This is absolutely critical if NeuralGCM is to be used as part of a climate model, as the manuscript seems to propose. At least suggesting a path forward here would be useful.

We have added to the discussion: “However, we will likely need alternative training strategies [9, 10] to learn important processes for climate with subtle impact on weather timescales, such as a cloud feedback.” and also: “For climate projection, NeuralGCM will need to be reformulated to enable coupling with other Earth system components (e.g., ocean, land), and integrating data on the atmospheric chemical composition (e.g., greenhouse gasses, aerosols). There are also research challenges common to current ML-based climate models [2], including the capability to simulate unprecedented climates (i.e., generalization), adhering to physical constraints, and resolving numerical instabilities and climate drift.”

We also write in the SI now: “In the future, NeuralGCM could improve its ability to generalize to different climates, even those it has not trained on, by using a mix of strategies. While climate-invariant methods [3] can help, training NeuralGCM on data obtained from simulations (e.g., of warmer climates) and incorporating physically-based models (e.g., radiation schemes) will likely be necessary.”

Line 994: what exactly do you mean by “temperature variation”?

We have now changed the terminology to temperature deviation, and as explained in the SI: “To facilitate efficient time integration of our models we split temperature T into a uniform reference temperature on each sigma level \bar{T}_σ and temperature deviations per level $T'_\sigma = T_\sigma - \bar{T}_\sigma$ ”

Line 1011: “receives an additional” -> “receives additional”

Done

Line 1089-1093: why the different configuration for different resolutions w.r.t the land/sea-ice embeddings?

The reason we use different embeddings for the NeuralGCM-2.8° compared to the NeuralGCM-1.4° and NeuralGCM-0.7° simulations is that when we trained our NeuralGCM-2.8° configurations with the land/sea-ice embeddings utilized by the NeuralGCM-1.4° and NeuralGCM-0.7°, the resulting checkpoints were less stable in AMIP runs. Consequently, we made several hyperparameter adjustments, one of which involved changing the inputs to the land/sea-ice embeddings, and this led to more stable AMIP simulations.

Section C.4: I don’t understand how the vertical/surface embedding networks interface with the EPD architecture discussed on lines 1066-1074. A schematic would be helpful.

Following the suggestion of the reviewer we have now added to the SI detailed a figure showing the structure of the learned physics module, and how the surface embedding network interface with the physics module (Fig 13).

Section D.2: how large are the corrections to linear interpolation from the decoder NN?

Following the comment of the reviewer, we have added Figs. 3,4 showing the differences between encoder/decoder components and direct interpolation between pressure and sigma levels is very small and most notable in areas like extrapolation to pressure levels below ground.

Line 1566: reference not defined

Corrected

Section G.7: how many trainable parameters are there for each resolution?

We now added table 1 (also added to the SI) showing that. The number of parameters are also shown below: Deterministic NeuralGCM-2.8°: 14.5 M, Deterministic NeuralGCM-1.4°: 18.3 M, Deterministic NeuralGCM-0.7°: 31.1 M, Stochastic (NeuralGCM-ENS): 11.5 M.

The main reason for the difference in the number of parameters we have learned is the location-specific embedding vector for each horizontal grid-point. For higher resolution models we need to learn more of these features. For the NeuralGCM-1.4° and NeuralGCM-0.7° models we have used an embedding of length 32 for every lat-lon, and for the NeuralGCM-ENS and NeuralGCM-2.8° models we have used an embedding of length 8. We now write this in the SI: “NeuralGCM models differ in parameter count primarily due to the location-specific embedding vector assigned to each horizontal grid point. Higher-resolution models require more parameters. Additionally, NeuralGCM-1.4° and NeuralGCM-0.7° models use 32-length embeddings, while NeuralGCM-ENS and NeuralGCM-2.8° models use 8-length embeddings.”

Figure H13: Is “NeuralGCM-HRES” the same as “NeuralGCM-0.7”? Didn’t see the “HRES” label used elsewhere.

Yes, this name was used accidentally and has been corrected.

Figure H17: where does the negative cloud liquid/ice come from? Dynamics or the ML? It is interesting how much NeuralGCM struggles to get the zero cloud regime correct.

The dynamics only advects the clouds, so we think that negative clouds are due to the learned physics tendencies.

Figure I24a: Interesting (and likely unrealistic) big drops in precipitable water for at least one ensemble members. Any idea of what the mechanism is?

We agree with this observation, and we currently do not have any idea what the mechanism is.

References

- [1] Mardani, M. *et al.* Residual diffusion modeling for km-scale atmospheric downscaling (2024).
- [2] Bretherton, C. S. Old dog, new trick: Reservoir computing advances machine learning for climate modeling. *Geophysical Research Letters* **50**, e2023GL104174.
- [3] Beucler, T. *et al.* Climate-invariant machine learning. *Science Advances* **10**, eadj7250 (2024).

- [4] Brenowitz, N. D., Beucler, T., Pritchard, M. & Bretherton, C. S. Interpreting and stabilizing machine-learning parametrizations of convection. *Journal of the Atmospheric Sciences* **77**, 4357–4375 (2020).
- [5] Watt-Meyer, O. *et al.* ACE: A fast, skillful learned global atmospheric model for climate prediction. *arXiv preprint arXiv:2310.02074* (2023).
- [6] Weyn, J. A., Durran, D. R. & Caruana, R. Improving data-driven global weather prediction using deep convolutional neural networks on a cubed sphere. *Journal of Advances in Modeling Earth Systems* **12**, e2020MS002109 (2020).
- [7] Han, Y., Zhang, G. J. & Wang, Y. An ensemble of neural networks for moist physics processes, its generalizability and stable integration. *Journal of Advances in Modeling Earth Systems* **15**, e2022MS003508 (2023).
- [8] Wang, P., Yuval, J. & O’Gorman, P. A. Non-local parameterization of atmospheric subgrid processes with neural networks. *Journal of Advances in Modeling Earth Systems* **14**, e2022MS002984 (2022).
- [9] Schneider, T., Lan, S., Stuart, A. & Teixeira, J. Earth system modeling 2.0: A blueprint for models that learn from observations and targeted high-resolution simulations. *Geophysical Research Letters* **44**, 12–396 (2017).
- [10] Schneider, T., Leung, L. R. & Wills, R. C. J. Opinion: Optimizing climate models with process-knowledge, resolution, and AI. *EGUsphere [preprint]* (2024). URL <https://egusphere.copernicus.org/preprints/2024/egusphere-2024-20/>.

Reviewer Reports on the First Revision:

Referees' comments (note that referee #2 references Nature Climate Change - we are going with the assumption that this is a typo!):

Referee #2 (Remarks to the Author):

The authors have diligently revised their manuscript, incorporating significant details about their models. In particular, the ablation and generalization studies help readers to better understand the NGCM's characteristics and performance. Additionally, the revised manuscript provides a more objective assessment of the model, including its limitations. All my previous concerns have been satisfactorily addressed. I believe the current version of the manuscript meets the high standards required for publication in Nature Climate Change. Given the rapid pace of advancements in this field, timely publication of this work would be beneficial for the community.

Referee #2 (Remarks on code availability):

The URL above is incorrect. <https://github.com/google-research/neuralgcm>

The code repositories (neuralgcm and dinosaur) are well-curated, including the quick-start notebooks.

Referee #3 (Remarks to the Author):

The authors have satisfied my concerns regarding the manuscript. I just note some minor issues found during reading the revised manuscript.

Lines 415-416: Should refer to Fig I47, not I49.

Figure 4d: typo in subplot title

Code availability: I'm not sure about Nature policy but I think the authors should archive their code (e.g. in a zenodo repository) because GitHub repos are not guaranteed to be long lived. Then a DOI can be provided in addition to the GitHub URL.

Lines 1207-1208: typo

Referee #3 (Remarks on code availability):

The link provided above is incorrect. But using the URL provided in the manuscript (<https://github.com/google-research/neuralgcm>) I was able to view and use the code.

In general, the code is of very high quality. It should be easy to use for advanced users, especially those with familiarity with cloud computing. In particular, I was able to run the coarsest resolution model in the example notebook on a Google Colab session with about half an hour of effort.

Author Rebuttals to First Revision:

Response to Referee #2

The authors have diligently revised their manuscript, incorporating significant details about their models. In particular, the ablation and generalization studies help readers to better understand the NGCM's characteristics and performance. Additionally, the revised manuscript provides a more objective assessment of the model, including its limitations. All my previous concerns have been satisfactorily addressed. I believe the current version of the manuscript meets the high standards required for publication in Nature Climate Change. Given the rapid pace of advancements in this field, timely publication of this work would be beneficial for the community. We thank the reviewer for his feedback which has helped us to improve the manuscript.

Referee #2 (Remarks on code availability):

The URL above is incorrect. <https://github.com/google-research/neuralgcm>

We submitted the incorrect URL for code resources along with the previous submission. The URLs for code in the manuscript are correct.

The code repositories (neuralgcm and dinosaur) are well-curated, including the quick-start notebooks.

Thanks for the feedback!

Response to Referee #3

The authors have satisfied my concerns regarding the manuscript. I just note some minor issues found during reading the revised manuscript.

We thank the reviewer for his feedback which has helped us to improve the manuscript.

Lines 415-416: Should refer to Fig I47, not I49.

Done. Thanks for catching this mistake!

Figure 4d: typo in subplot title

We have corrected the typo. Thanks for catching it!

Code availability: I'm not sure about Nature policy but I think the authors should archive their code (e.g. in a zenodo repository) because GitHub repos are not guaranteed to be long lived. Then a DOI can be provided in addition to the GitHub URL.

Done. All code has been archived in Zenodo.

Lines 1207-1208: typo

We have double checked the equation and could not find any errors. Possibly the referee expected to see the term in the exponential squared, but the term is correct.

Referee #3 (Remarks on code availability):

The link provided above is incorrect. But using the URL provided in the manuscript (<https://github.com/google-research/neuralgcm>) I was able to view and use the code.

Currently, the URL in the manuscript is correct and leads to the correct github repository

In general, the code is of very high quality. It should be easy to use for advanced users, especially those with familiarity with cloud computing. In particular, I was able to run the coarsest resolution model in the example notebook on a Google Colab session with about half an hour of effort.

Thanks for the feedback!

Notable text changes since previous submission

1. *First paragraph*: For both weather and climate, our approach offers orders of magnitude computational savings over conventional GCMs, **although our model does not extrapolate to substantially different future climates.**
2. *Neural General Circulation Models*: our models are not accurate for multi-day prediction **or stable for long rollouts** early in training
3. *Generalizing to unseen data*: Physically consistent weather models should still perform well for weather conditions for which they were not trained. **We expect that NeuralGCM may generalize better than ML-only atmospheric models, because NeuralGCM employs neural networks that act locally in space, on individual vertical columns of the atmosphere.** ~~To test generalization in the context of weather~~ **To explore this hypothesis,** we compare versions of NeuralCGM0.7^o and GraphCast trained through 2017 on five years of weather forecasts beyond the training period (2018-2022) in SI Fig. 36
4. *Figure 4*: We have shortened the caption to fit below the limit of 300 words (previously it was at ~370 words).